# Elevated extracellular matrix protein 1 in circulating extracellular vesicles supports breast cancer progression under obesity conditions

Keyang Xu [1], Ai Fu[2], Zhaoyi Li[2], Liangbin Miao[2], Zhonghan Lou[2], Keying Jiang[1], Condon Lau[3], Tao Su[4], Tiejun Tong [5], Jianfeng Bao[2] ✉, Aiping Lyu [1,6] ✉ & Hiu Yee Kwan [1,6,7] ✉

The cargo content in small extracellular vesicles (sEVs) changes under pathological conditions. Our data shows that in obesity, extracellular matrix protein 1 (ECM1) protein levels are significantly increased in circulating sEVs, which is dependent on integrin-β2. Knockdown of integrin-β2 does not affect cellular ECM1 protein levels but significantly reduces ECM1 protein levels in the sEVs released by these cells. In breast cancer (BC), overexpressing ECM1 increases matrix metalloproteinase 3 (MMP3) and S100A/B protein levels. Interestingly, sEVs purified from high-fat diet-induced obesity mice (D-sEVs) deliver more ECM1 protein to BC cells compared to sEVs from control diet-fed mice. Consequently, BC cells secrete more ECM1 protein, which promotes cancer cell invasion and migration. D-sEVs treatment also significantly enhances ECM1-mediated BC metastasis and growth in mouse models, as evidenced by the elevated tumor levels of MMP3 and S100A/B. Our study reveals a mechanism and suggests sEV-based strategies for treating obesity-associated BC.

Small extracellular vesicles (sEVs) are specialized vesicles enclosed by lipid bilayer, with a particle diameter ranging from 50 to 200 nm[1]. These vesicles contain proteins, nucleic acids, and lipids[2], and play a crucial role in mediating communication between neighboring cells or acting on distant recipient targets. In the context of breast cancer (BC), sEVs have been implicated in promoting metastasis[3,4]. BC cells release sEVs that stimulate angiogenesis in the microenvironment through the circHIPK3/miR-124-3p/MTDH axis[5]. The content of these sEVs can also contribute to the formation of pre-metastatic niches[6], facilitating the proliferation of metastatic tumor cells[7]. For instance, sEVs derived from invasive BC cells and glioma cells contain HSP90α, which enhances cancer cell motility by converting plasminogen into plasmin[8]. However, whether sEVs contribute to the enhanced BC metastasis and growth under obesity conditions has not been explored.

Clinical studies have consistently shown that obesity has a significant impact on the development of BC. BC patients with obesity tend to have larger tumors, more advanced disease stage at diagnosis,

[1]Centre for Cancer & Inflammation Research, School of Chinese Medicine, Hong Kong Baptist University, Hong Kong, China. [2]Hangzhou Xixi Hospital, Zhejiang Chinese Medical University, Hangzhou, China. [3]Department of Physics, City University of Hong Kong, Hong Kong, China. [4]International Institute for Translational Chinese Medicine, School of Pharmaceutical Science, Guangzhou University of Chinese Medicine, Guangzhou, China. [5]Department of Mathematics, Hong Kong Baptist University, Hong Kong, China. [6]Institute of Systems Medicine and Health Sciences, Hong Kong Baptist University, Hong Kong, China. [7]Institute of Research and Continuing Education, Hong Kong Baptist University, Shenzhen, China. ✉e-mail: zjbjf1972@aliyun.com; aipinglu@hkbu.edu.hk; hykwan@hkbu.edu.hk

higher rates of metastasis, and increased risk of distant recurrence compared to BC patients without obesity[9–14]. Obesity-related variables such as body mass index (BMI) and weight exert significant adverse univariable associations with BC distant recurrence over time (BMI is modeled quadratically, for the quartiles 4 vs. quartile 2: hazard ratio, 1.40; 95% CI, 1.07 to 1.82 for distant disease-free survival; P <0.014; and hazard ratio, 1.50; 95% CI, 1.16 to 1.93; P <0.001 for overall survival)[15]. BC women with obesity also have a significantly higher risk of all-cause mortality and BC-specific mortality compared to women of normal weight[13]. These effects have been observed in both premenopausal and postmenopausal BC patients[16–18].

While the dysfunction of adipocytes is found to underlie the enhanced development of BC in obesity, other pathological conditions may also contribute. For instance, BC cells can induce the secretion of matrix metalloproteinase-11 from adjacent adipocytes, leading to the dedifferentiation of adipocytes into fibroblast-like cells that support BC cell invasion[19,20]. Adipocytes themselves release interleukin-6, which promotes the invasion and migration of BC cells in the tumor microenvironment[21]. Various mechanisms may contribute to the enhanced growth and metastasis of BC under obesity conditions. The cargo contents of sEVs can be changed in pathological conditions[22,23], which contribute to the progression of the underlying pathological conditions[24]. Among the various cargo components, sEV proteins are considered stable and have a longer half-life[25]. They can directly interact with target signaling molecules[26], thereby mediating biological or pathological effects.

In this study, our focus was to investigate whether the protein content of sEVs was changed under obesity conditions that underlie the enhanced BC development. Using iTRAQ-based quantitative proteomics, 4D-label free quantitative proteomics, and multiple reaction monitoring-mass spectrometry (MRM-MS), we found that obesity led to an increase in the levels of extracellular matrix protein 1 (ECM1) in the circulating sEVs of human subjects with obesity and high-fat diet-induced obesity (DIO) mouse models. Loading of ECM1 protein into sEVs was associated with integrin-β2. The levels of integrin-β2 were also elevated in macrophages and adipocytes under obesity conditions. Interestingly, circulating sEVs purified from DIO mice (D-sEVs) were found to deliver more ECM1 protein to BC cells compared to sEVs from control diet mice. The roles of ECM1 protein in the sEVs in promoting BC growth and metastasis were evidenced in different mouse models.

## Results

### Obesity increases ECM1 protein level in the circulating sEVs of human subjects with obesity and DIO mouse model

In our study, we obtained circulating sEVs from the plasma of 96 human subjects with obesity or overweight (OB) and 48 normal weight healthy subjects (NC). Transmission electron microscopy (TEM) and nanoparticle tracking analysis (NTA) were done to examine the diameters of the purified sEVs (Fig. 1a). The authenticity of the sEVs was also examined by the presence of sEV markers CD9, TSG101 and CD63 (Supplementary Figs. S1a, b and c). Absence of endoplasmic reticulum calnexin (Supplementary Fig. S1d) suggests little or no contamination of vesicles in the samples[27].

To examine the protein profiles of the sEVs, we performed proteomics analysis. We identified peptides that had a detection rate of above 60% in each group for further analysis. These peptides were subjected to variable statistical analysis, followed by bioinformatics analysis and data mining.

Our analysis revealed that the sEVs from patients with obesity or overweight exhibited differential expressions of proteins compared to the sEVs from healthy normal weight subjects. Specifically, we identified 86 upregulated proteins and 15 downregulated proteins in the sEVs of patients with obesity or overweight, as shown in Supplementary Data 1. To visually represent these differentially expressed

proteins (DEPs), we generated a volcano plot (Fig. 1b) and a heatmap (Fig. 1c). The volcano plot provides a graphical representation of the fold change and statistical significance of the DEPs, while the heatmap displays the expression patterns of the DEPs across the different groups. KEGG pathway analysis showed that these DEPs were mostly involved in the complement and coagulation cascades, bacterial infection, and ECM-receptor interactions (Fig. 1d). Among the top four highlighted signaling pathways, ECM-receptor interaction contributes to the metastasis of BC[28].

In addition to the human study, we also investigated the protein profiles of plasma sEVs in DIO and CD mice. TEM and NTA (Fig. 1e) were done to examine the size of the sEVs purified from these mice. Authenticity of the purified sEVs was confirmed by the presence of CD63 and CD81 (Supplementary Fig. S1e). The proteomics data showed that a total of 38 proteins were upregulated, and 68 proteins were downregulated in the sEVs of the DIO mice when compared to CD mice (Supplementary Data 2). GO analysis revealed that these DEPs were mainly involved in molecular binding (Fig. 1f).

Among the DEPs in the human and mouse samples, 15 of them overlapped (Fig. 1g and Table 1). Notably, CPN2 (carboxypeptidase N Subunit 2), F5 (coagulation factor V), ECM1 (extracellular matrix protein 1), KNG1 (kininogen 1), FN1 (fibronectin 1), and GPLD1 (glycosyl-phosphatidylinositol-specific phospholipase D1) were found to be elevated in both human and mouse sEVs under obesity conditions. In the mouse samples, it is particularly noteworthy that the sEVs derived KNG1 protein level in sEVs exhibited a significant upregulation of 5.65-fold, followed by a 2.63-fold upregulation in the ECM1 protein level under obesity conditions (Table 1).

To validate and quantify the differentially expressed proteins (DEPs) in human plasma sEVs, we employed multiple reaction monitoring (MRM) mass spectrometry. MRM utilizes a triple quadrupole mass spectrometer to detect the mass spectral response signals of both the parent ion and product ion of the target molecule, providing highly reproducible and sensitive qualitative and quantitative information for analysis. Supplementary Fig. S2a demonstrates the stable retention time distribution; Supplementary Fig. S2b indicates the qualitative stability of repeated tests and minimal systemic errors in the samples. The quantitative results of the detected proteins in the quality control (QC) samples showed consistent quantitative signal values with coefficients of variation (CVs) all below 10%, indicating the stability of the instrument signal response (Supplementary Fig. S2c). The selected peptides used for quantification were listed in Supplementary Data 3. The MRM data validated a significant elevation of ECM1, but not KNG1, in the sEVs of human subjects with obesity or overweight (Supplementary Data 4). Additionally, we also examined the ECM1 protein level in the mouse sEVs using Western blot analysis. As shown in Fig. 1h, ECM1 protein level was significantly elevated in the sEVs of DIO mice (D-sEVs) compared to those in control diet (CD) mice (C-sEVs). Furthermore, we found that the majority of ECM1 protein in circulation was transferred via sEVs, as relatively less ECM1 protein was detected in the sEV-depleted plasma of the mice (Fig. 1i). Collectively, our data strongly suggest that obesity leads to increased ECM1 protein level in the circulating sEVs.

### Loading of ECM1 protein into sEVs is associated with integrin-β2

Integrins are transmembrane receptors that mediate the interactions between the cytoskeleton and extracellular matrix. Integrins comprise an α and a β subunit[29,30]. Integrin-β2 is coded by the gene ITGB2 and is found in extracellular vehicles[31]. A recent study also reported that integrin-β2 bound to ECM1 via the Gly-Pro-Arg (GPR) motif on the protein[32].

Our proteomics analysis revealed that integrin-β2 protein was detected in the circulating sEVs of human subjects, and its expression level was found to be elevated by 2.78-fold under obesity conditions

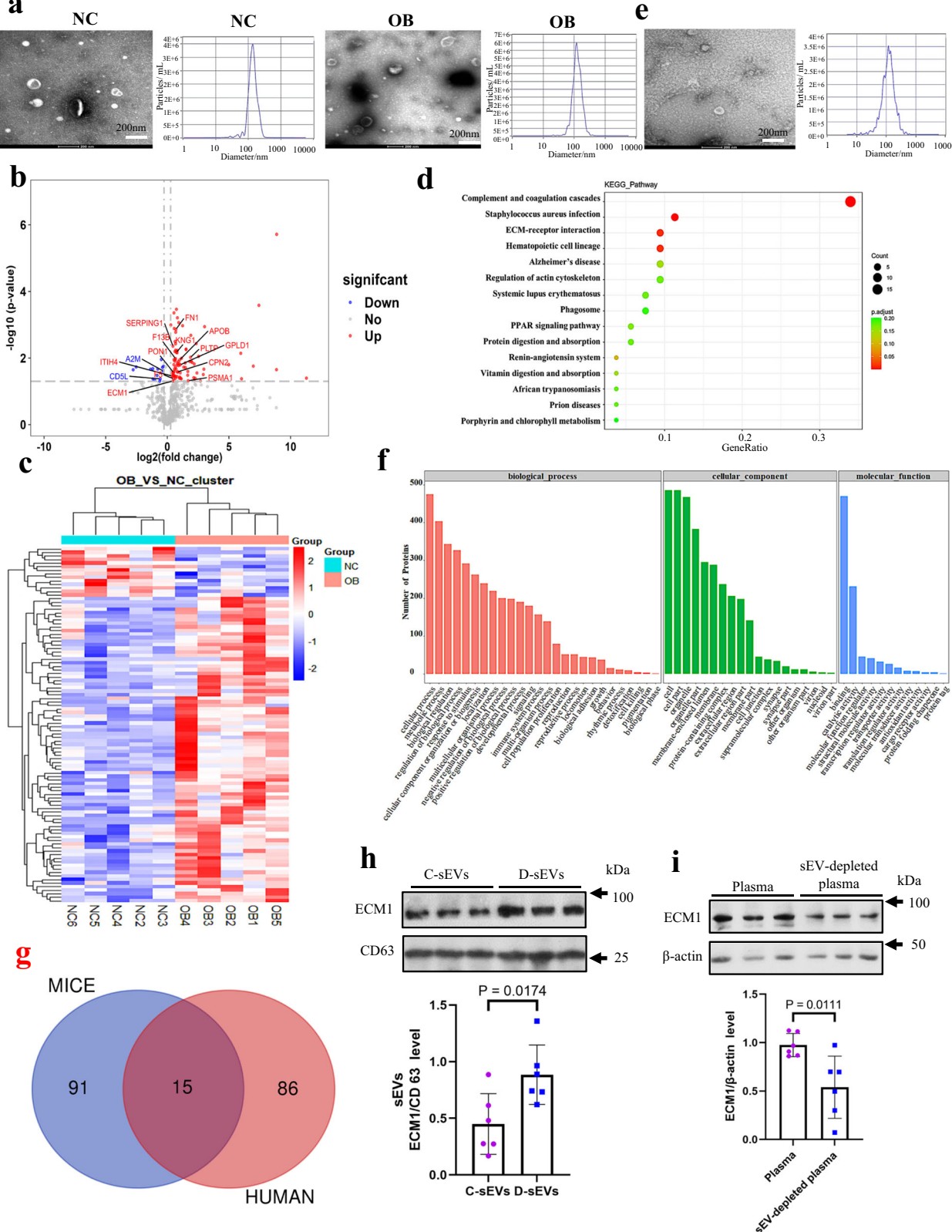

(Supplementary Data 1). Additionally, we also observed an increased expression level of integrin-β2 in the plasma sEVs of DIO mice (Fig. 2a). By analyzing single-cell specific transcriptome profiles, we found that ECM1 was highly expressed in fibroblasts and various cell types (Fig. 2b). Further single cell sequencing analysis focusing on peripheral blood mononuclear cells (PBMC) revealed that ECM1 was prominently expressed in macrophages (Fig. 2c and Supplementary Fig. S3a). Interestingly, macrophages also expressed integrin-β2 (Fig. 2d and Supplementary Fig. S3b).

To investigate the role of integrin-β2 in the loading of ECM1 protein into sEVs, we utilized RAW264.7 cells as a model system. We purified the sEVs from RAW264.7 cells and performed transmission

**Fig. 1 | Obesity increases ECM1 protein levels in the circulating sEVs of the human subjects with obesity or overweight and high-fat diet-induced obesity (DIO) mouse models. a** Transmission electron microscopy (TEM) and nanoparticle tracking analysis (NTA) of the circulating sEVs purified from the plasma of normal weight healthy human subjects (NC) and human subjects with obesity or overweight (OB). **b** Volcano plot, **c** heatmap showing the differential expressed proteins (DEPs) in the sEVs. **d** Pathway enrichments analysis of the DEPs in the sEVs. **e** TEM and NTA of the circulating sEVs purified from plasma of the mouse models. **f** GO analysis of the DEPs in the sEVs of the mouse models. **g** Overlapping DEPs in the

human and mouse sEVs. **h** ECM1 protein levels in the circulating sEVs in CD mice (C-sEVs) and DIO mice (D-sEVs). **i** ECM1 protein levels in the plasma and sEVs-depleted plasma of DIO mice. Shown is the mean ± SD; two-sided unpaired *t*-test for (**b**, **d**, **h**, **i**); *n* = 3 independent experiments for (**a**, **e**); *n* = 6 mice in each group; *p* values were indicated in graphs. C-sEVs, circulating sEVs in control diet mice; D-sEVs, circulating sEVs in high-fat diet-induced obesity mice; sEV-depleted, sEVs-depleted plasma in high-fat diet-induced obesity mice; ECM1 extracellular matrix protein 1. Source data are provided in Source Data file.

## Table 1 | Differentially expressed proteins (DEPs) in the circulating sEVs

| Overlaping | Mice Accession | Trend | BT: BC | P value | Human Accession | Trend | OB: NC | P value |
|---|---|---|---|---|---|---|---|---|
| PON1 | P52430 | Down | 0.416869 | 0.072844 | P27169 | Up | 1.410720 | 0.026810 |
| SERPING1 | P97290 | Down | 0.679204 | 0.246092 | P05155 | Up | 1.373380 | 0.004385 |
| F13B | Q3UER0 | Down | 0.501187 | 0.000005 | P05160 | Up | 1.584380 | 0.010348 |
| CPN2 | Q9DBB9 | Up | 1.819701 | 0.082274 | P22792 | Up | 1.759430 | 0.026131 |
| APOB | E9Q1Y3 | Down | 0.343558 | 0.000955 | P04114 | Up | 2.001660 | 0.011371 |
| F5 | O88783 | Up | 2.558586 | 0.011448 | P12259 | Up | 1.441020 | 0.042143 |
| PSMA1 | Q3TS44 | Down | 0.549541 | 0.036935 | P25786 | Up | 3.236740 | 0.048003 |
| A2M | Q6GQT1 | Down | 0.564937 | 0.050095 | P01023 | Down | 0.797260 | 0.024444 |
| ITIH4 | E9Q5L2 | Down | 0.549541 | 0.008112 | Q14624 | Up | 1.310230 | 0.034684 |
| CD5L | Q9QWK4 | Down | 0.549541 | 0.020155 | O43866 | Down | 0.671800 | 0.042232 |
| ECM1 | Q3TXB7 | Up | 2.630268 | 0.000273 | Q16610 | Up | 1.357100 | 0.049087 |
| KNG1 | A0A0R4J038 | Up | 5.649370 | 0.021724 | P01042 | Up | 1.673730 | 0.006762 |
| FN1 | A0A087WSN6 | Up | 1.584893 | 0.045708 | P02751 | Up | 1.626480 | 0.001375 |
| PLTP | Q3UFS5 | Down | 0.373250 | 0.122686 | P55058 | Up | 3.696800 | 0.009497 |
| GPLD1 | Q7TNZ4 | Up | 2.269865 | 0.010958 | P80108 | Up | 2.055300 | 0.016015 |

Comparison of the differentially expressed proteins (DEPs) in the circulating sEVs purified from the plasma of the human subjects with obesity or overweight (OB) and normal weight healthy human subjects (NC), and the DEPs of the circulating sEVs purified from the plasma of the control diet-fed mice (BC) and high fat diet-induced obesity mice (BT). Two-sided unpaired *t*-test are used for statistical analysis.

electron microscopy (TEM) studies to examine their size and morphology (Supplementary Fig. S4a). Additionally, nanoparticle tracking analysis (NTA) was conducted to determine the size distribution of the purified sEVs (Supplementary Fig. S4b). The knockdown of integrin-β2 in RAW264.7 cells was firstly confirmed at gene and protein levels (Supplementary Figs. S5a and 2e). Notably, the knockdown of integrin-β2 did not significantly affect the levels of ECM1 protein within the macrophages (Supplementary Figs. S5b and 2e). However, it reduced the ECM1 protein levels in the sEVs released by these macrophages (Fig. 2f). Furthermore, we examined the levels of other sEV proteins that lack the GPR motif. Our results demonstrated that the knockdown of integrin-β2 in macrophages did not significantly impact the levels of intracellular adhesion molecule 1 (ICAM1) and glutathione peroxidase 3 (GPX3) within the sEVs (Fig. 2g). Besides, we also investigated the effects of integrin-β2 overexpression in RAW264.7 cells. Interestingly, the overexpression of integrin-β2 did not significantly alter the levels of ECM1 protein within the macrophages (Fig. 2h). However, it significantly increased both ECM1 and integrin-β2 protein levels in the sEVs released by these macrophages (Fig. 2i). Taken together, these findings strongly suggest that integrin-β2 plays a crucial role in the loading of ECM1 protein into sEVs.

## Integrin-β2 protein levels are elevated in the monocytes and adipocytes under obesity conditions

If the enhanced loading of ECM1 protein into sEVs is facilitated by integrin-β2, integrin-β2 expression should be elevated in the cells under obesity conditions. Indeed, under obesity conditions, integrin-β2 protein level in monocytes was significantly increased (Fig. 3a); while the protein level of ECM1 was not significantly changed (Fig. 3a). Interestingly, both integrin-β2 and ECM1 protein levels in the sEVs

released by the monocytes in DIO mice were significantly increased when compared to those in CD mice (Fig. 3b).

In addition to monocytes, we also examined the expression of ECM1 and integrin-β2 in adipocytes. Our results showed that adipocytes indeed express both ECM1 (Supplementary Fig. S6a and b) and integrin-β2 (Supplementary Fig. S6c and d). Interestingly, we found that integrin-β2 protein level was elevated in the subcutaneous adipose tissues of DIO mice (Fig. 3c). Furthermore, the sEVs released from these adipose tissues had elevated integrin-β2 and ECM1 protein levels (Fig. 3d).

Our data not only demonstrate the association between enhanced sEV ECM1 protein levels and integrin-β2 expression, but also suggest that monocytes/macrophages and adipocytes may serve as potential donor cells releasing sEVs with elevated ECM1 protein levels under obesity conditions.

## The overexpression of ECM1 in BC cells leads to an increase in MMP3 and S100A/B, both of which play a crucial role in the development of BC

Next, we utilized Virtual Flow analysis and prognosis analysis to investigate the impact of ECM1 on the growth and metastasis of BC. Our findings revealed that ECM1 was significantly associated with epithelial-mesenchymal-transition (EMT) markers (Supplementary Fig. S7a), angiogenesis (Supplementary Fig. S7b), P53 (Supplementary Fig. S7c), apoptosis (Supplementary Fig. S7d), and transforming growth factor beta (TGF-β) signaling pathways (Supplementary Fig. S7e).

In order to further confirm the pro-metastatic effects of ECM1, we conducted an experiment where ECM1 was overexpressed in human breast cancer cells (HCC1806) (Supplementary Fig. S7f). Our results demonstrated that the overexpression of ECM1 significantly enhanced

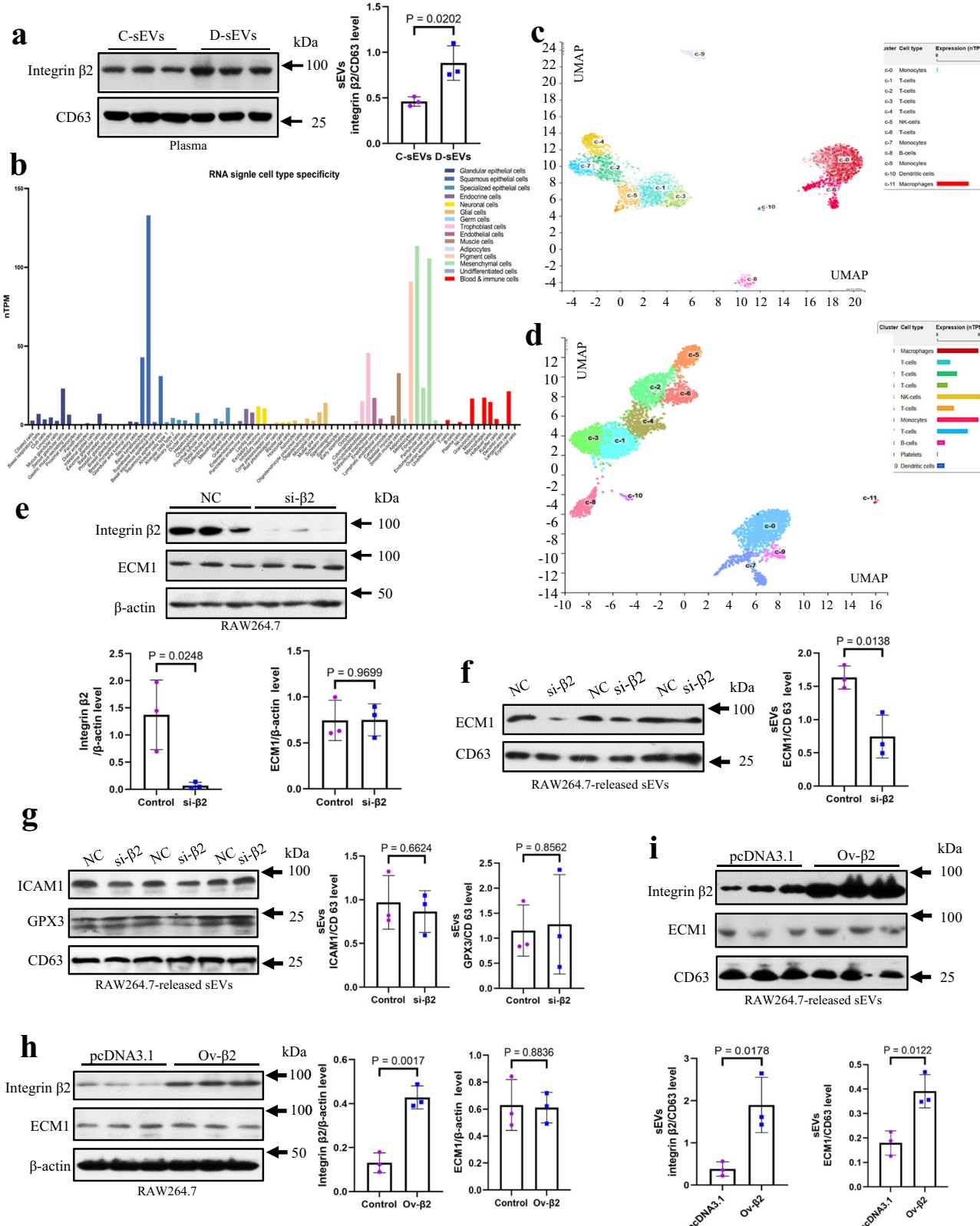

the invasion (Supplementary Fig. S7g) and migration (Supplementary Fig. S7h) capabilities of the BC cells. Interestingly, we also found that the expression levels of MMP3 and S100A/B were significantly increased in the ECM1-overexpressed HCC1806 cells (Supplementary Fig. S7i). Prognosis analysis further supported the involvement of ECM1, MMP3, and S100A2 in the development of BC, as indicated by their elevated levels in human tumor biopsies (Supplementary Fig. S7j).

**Circulating sEVs from DIO mice deliver more ECM1 protein to the cancer cells, which in turn secret more ECM1 that acts on the cancer cells to increase invasion and migration**

While we have demonstrated the pro-metastatic effects of ECM1, it remained unclear whether the ECM1 protein present in sEVs could be transferred to BC cells and influence their metastatic potential.

**Fig. 2 | Loading of ECM1 protein into sEVs is associated with integrin-β2.**
**a** Integrin-β2 protein expressions in the circulating sEVs of high-fat diet-induced obesity mice (D-sEVs) and control diet mice (C- sEVs). **b** Single cell specific transcriptome profiles showing ECM1 expressions in different cell types. **c** Single cell sequencing analysis showing ECM1 expressions in peripheral blood mononuclear cells (PBMC) including macrophages. **d** Single cell type specificity showing integrin-β2 expressions in PBMC including macrophages. **e** Integrin-β2 protein levels in the integrin β2-knockdown and control RAW264.7 cells. **f** ECM1 and (**g**) ICAM1 and

GPX3 protein levels in the sEVs released by integrin-β2-knockdown and control RAW264.7 cells. Integrin-β2 protein levels (**h**) in integrin-β2-overexpressed and control RAW264.7 cells and (**i**) in the sEVs released by these cells. Shown is the mean ± SD; two-sided unpaired $t$-test for (**a**, **e** to **i**); n = 3 independent experiments; $p$ values are indicated in graphs. si-β2 integrin-β2-knockdown cells, NC control cells, ICAM-1 intercellular cell adhesion molecule-1, GPX3 glutathione peroxidase 3, ECM1 extracellular matrix protein 1, CD control diet, DIO high-fat diet-induced obesity. Source data are provided in Source Data file.

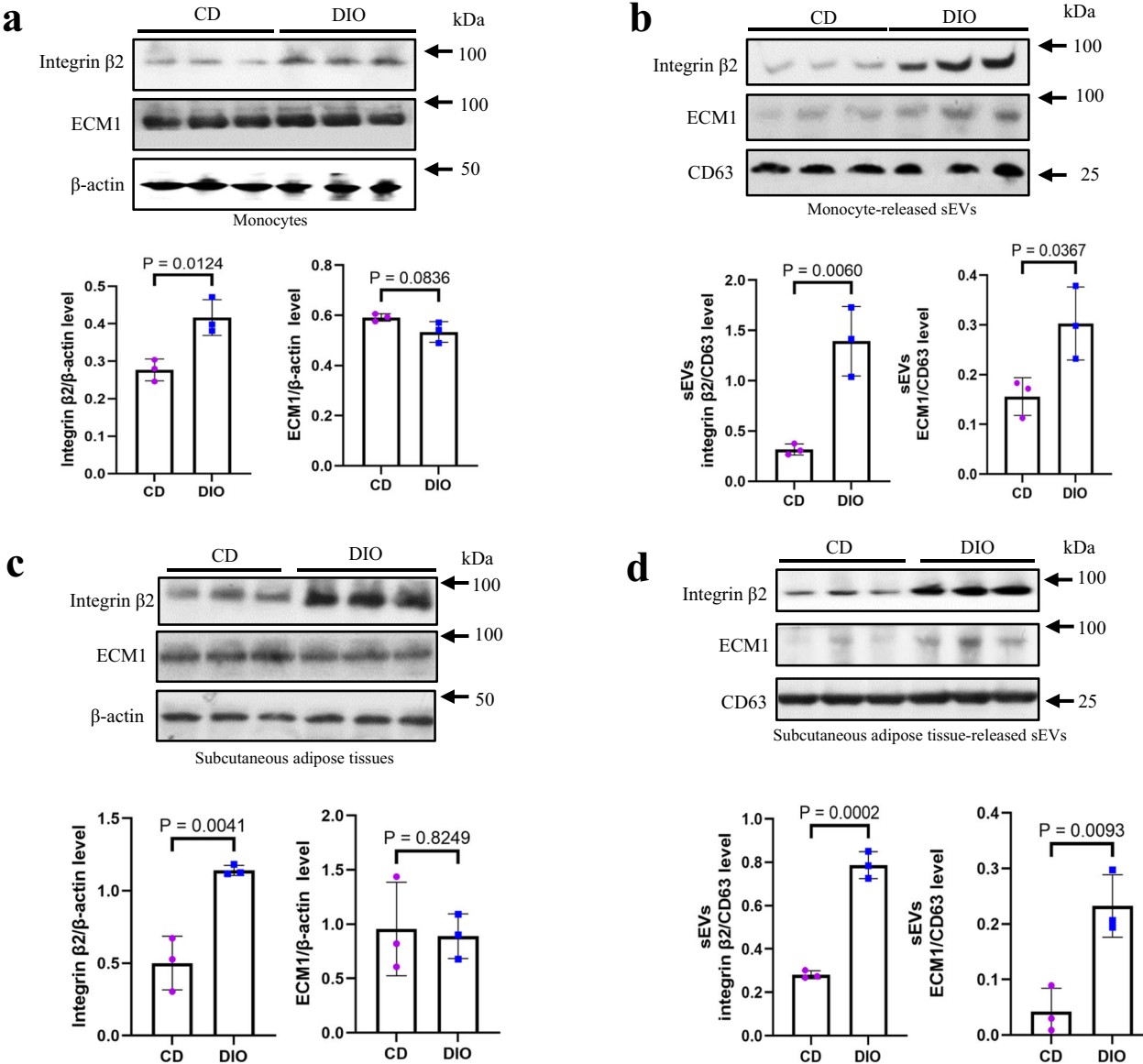

**Fig. 3 | Integrin-β2 and ECM1 protein levels are elevated in the monocyte-released and adipose tissue-released sEVs under obesity conditions. a** Integrin-β2 and ECM1 protein levels in the blood monocytes in CD and DIO mice. **b** Integrin-β2 and ECM1 protein levels in the sEVs released by the monocytes in CD and DIO mice. **c** Integrin-β2 and ECM1 protein levels in the subcutaneous adipose tissues in CD and DIO mice. **d** Integrin-β2 and ECM1 protein levels in the sEVs released by the subcutaneous adipose tissues in CD and DIO mice. Shown is the mean ± SD; two-sided unpaired $t$-test for (**a** to **d**); n = 3 mice in each group; $p$ values are indicated in graphs. CD control diet, DIO high-fat diet-induced obesity. Source data are provided as a Source Data file.

Here, we isolated sEVs from C57BL/6 mice and labeled them with PKH67. These labeled sEVs were then used to treat E0771 cells that were cultured in medium prepared with sEV-depleted fetal bovine serum (FBS). Our findings revealed that the E0771 cells could efficiently uptake the PKH67-labeled sEVs (Fig. 4a). Considering that not all proteins within the sEVs can be transferred to recipient cells[33], we

proceeded to examine whether the treatments would lead to an increase in cellular ECM1 protein levels. Compared to the cells treated by sEVs purified from CD mice (C-sEVs), D- sEVs s treatments significantly increased ECM1 protein level in E0771 cells as demonstrated by the immunofluorescence staining (Fig. 4b) and Western blot analysis (Fig. 4c), while ECM1 mRNA levels remained unchanged (Fig. 4d).

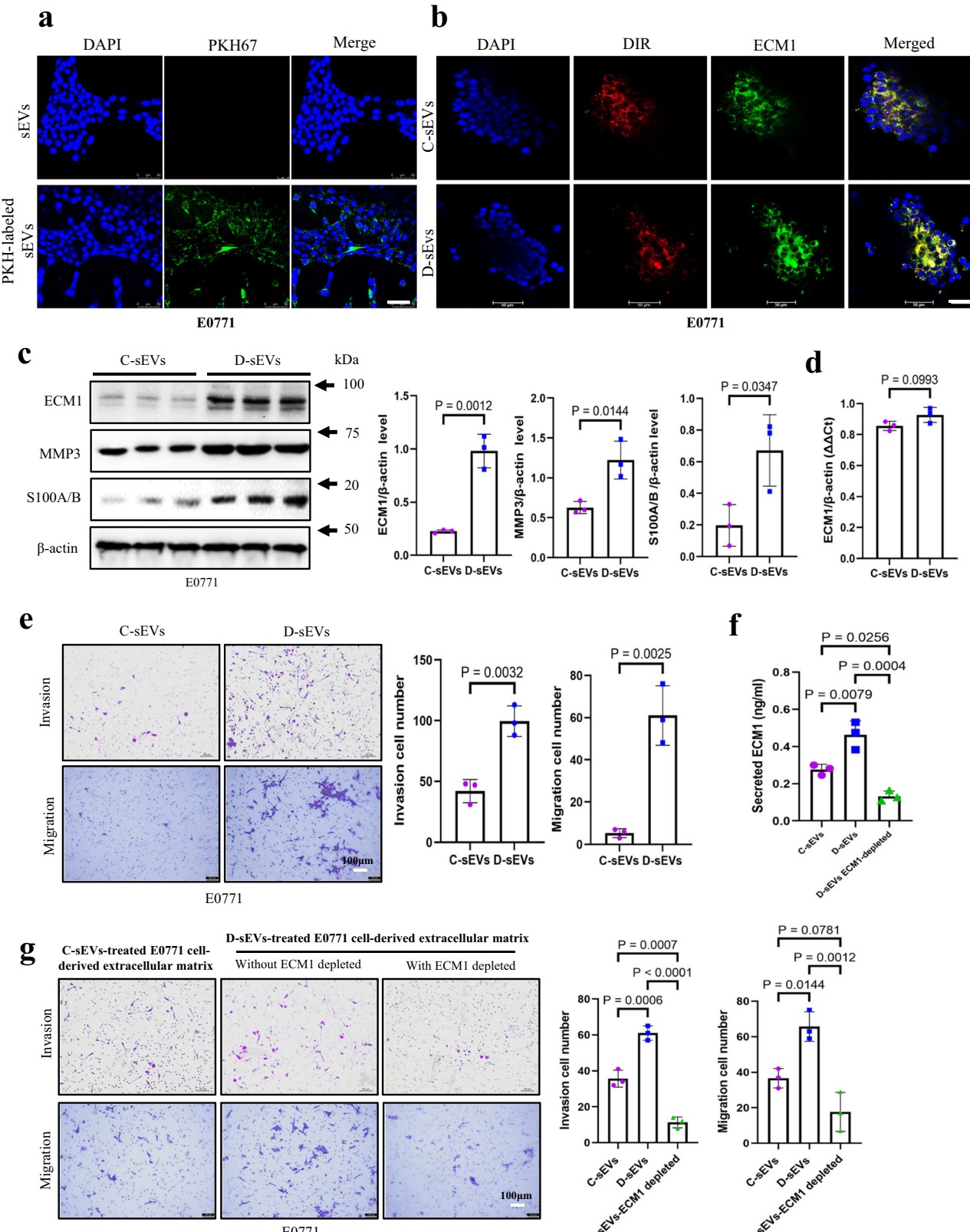

The data suggests that the elevated ECM1 protein level in the E0771 cells after treatments is not due to the transfer of RNA from sEVs. Subsequently, we investigated the functional implications of the transferred ECM1 protein in BC cells. Following treatment with D-sEVs, we observed a significant increase in the levels of MMP3 and S100A/B proteins in E0771 cells (Fig. 4c). These findings were consistent with

our previous observations where overexpression of ECM1 in BC cells resulted in enhanced MMP3 and S100A/B protein levels (Supplementary Fig. S7i). Importantly, D- sEVs treatments also significantly increased the invasive and migratory potential of the E0771 cells (Fig. 4e). Furthermore, we also treated the 4T1 cells with the circulating sEVs purified from the plasma of HFD-fed BALB/c mice (H-sEVs).

**Fig. 4 | Circulating sEVs from DIO mice deliver more ECM1 protein to the cancer cells, which in turn secret more ECM1 that acts on the cancer cells to increase invasion and migration. a** Uptake of PKH67-labeled sEVs by E0771 cells. Expression of ECM1 protein in sEVs-treated E0771 cells shown by (**b**) immunofluorescence staining and (**c**) Western blot analysis. **d** Expression of ECM1 mRNA levels in the sEVs-treated E0771 cells. **e** Invasion and migration of sEVs-treated E0771 cells after C-sEVs or D-sEVs treatments. **f** ECM1 protein levels in the extracellular matrix derived from the C-sEV-treated or D-sEV-treated E-0771 cells, and ECM1 protein level in the extracellular matrix after ECM1 was depleted by immunoprecipition

(D-sEVs ECM1-depleted). **g** Invasion and migration of E0771 cells following treatments with extracellular matrix derived from E-0771 cells treated with C-sEVs, D-sEVs, or ECM1-depleted ECM. Shown is the mean ± SD; two-sided unpaired *t*-test for (**c** to **e**) and one-way ANOVA with Tukey's multiple comparison test for (**f** to **g**); n = 3 independent experiments; *p* values are indicated in graphs. C-sEVs, circulating sEVs in CD mice; D-sEVs, circulating sEVs in high fat diet-induced obesity mice; ECM1 extracellular matrix protein 1; MMP3 matrix metallopeptidase 3. Source data are provided as a Source Data file.

Similar to the previous findings, treatment with H-sEVs led to an increase in ECM1 protein levels in 4T1 cells (Supplementary Fig. S8a and b), while the ECM1 mRNA levels remained unchanged (Supplementary Fig. S8c). Additionally, the levels of MMP3 and S100A/B proteins were also elevated in the cells following H-sEVs treatments (Supplementary Fig. S8b). Notably, H-sEVs treatments resulted in increased invasion and migration of 4T1 cells (Supplementary Fig. S8d).

Given ECM1 is a secretory protein, we next examined whether increased cellular ECM1 protein levels would enhance ECM1 secretion. Fig. 4f demonstrates that after D-sEVs treatments, E0771 cells secreted a significantly higher amount of ECM1 protein into the extracellular matrix compared to the cells treated with C-sEVs. Moreover, the extracellular matrix collected from D-sEVs-treated cells significantly promoted cell invasion and migration compared to the matrix derived from C-sEVs -treated cells (Fig. 4g). To investigate whether the elevated invasion and migration were indeed a result of the increased ECM1 protein in the extracellular matrix, we used immunoprecipitation to selectively deplete ECM1 protein from the matrix (Fig. 4f) before using it for treatments. As shown in Fig. 4g, after ECM1 was depleted from the extracellular matrix derived from the D-sEVs-treated cells, the treatments failed to increase cell invasion and migration. These findings collectively suggest that D-sEVs increase the delivery of ECM1 protein to cancer cells, and these cells increase ECM1 secretion. The secreted ECM1 acts on the cancer cells to promote invasion and migration.

### Obesity fails to increase cancer growth and metastasis in Rab27a knockout mouse model

To investigate the role of sEVs in promoting BC growth and metastasis under obesity conditions, we generated a mouse model with a knockout of Rab27a (B6/J-Rab27a-Cas9-KO). Rab27a is crucial for sEVs secretion, and constitutive knockout of Rab27a has been shown to reduce sEVs secretion[34]. Genotyping of the B6/J-Rab27a-Cas9-KO mice confirmed the successful knockout of Rab27a (Supplementary Fig. S9a). According to the Human Protein Atlas database, Rab27a is expressed in various tissues and organs, including the heart, liver, spleen, lung, kidney, brain, and adipose tissues. Western blot analysis validated the knockout of Rab27a protein in these major organs and tissues (Supplementary Fig. S9b). Both the B6/J-Rab27a-Cas9-KO mice and the isogenic control C57BL/6 J mice were fed HFD to induce obesity (Supplementary Fig. S9c). Subsequently, E0771 cells were inoculated into these mice. Interestingly, in the B6/J-Rab27a-Cas9-KO mice, HFD feeding failed to increase BC growth (Supplementary Fig. S9d, S9e, and S9f) and cancer lung metastasis (Supplementary Fig. S9g). Furthermore, there were no significant differences in tumor ECM1 protein levels between the DIO and control diet-fed B6/J-Rab27a-Cas9-KO mice (Supplementary Fig. S9h).

In contrast, in the E0771-bearing C57BL/6 mouse model, obesity not only led to an increase in ECM1 protein levels within sEVs (Fig. 1i), but it also resulted in a significant elevation of tumor ECM1 protein levels, as well as the downstream proteins MMP3 and S100A/B, compared to CD mice (Supplementary Fig. S10a). Furthermore, the DIO mice had larger tumors (Supplementary Fig. S10b, c, and d) and enhanced lung metastasis (Supplementary Fig. S10e). Similar results were observed in the 4T1-bearing BALB/c mouse model, where HFD

feeding significantly increased ECM1 protein levels in the sEVs (Supplementary Fig. S11a), as well as ECM1, MMP3, and S100A/B protein levels in the tumor tissues (Supplementary Fig. S11b, c). HFD feeding also resulted in increased tumor size (Supplementary Fig. S11d and e), tumor weight (Supplementary Fig. S11f), and lung metastasis (Supplementary Fig. S11g). Taken together, the findings highlight the importance of sEVs in promoting BC growth and metastasis under obesity, and the impact of obesity-induced changes in the ECM1 protein level in the sEVs.

### Injection of H-sEVs increases tumor ECM1 levels, cancer growth and metastasis in CD-fed 4T1-bearing mouse model

Next, we investigated the role of H-sEVs in promoting BC growth and metastasis in vivo. We first examined whether sEVs would accumulate at the tumor site. We established a mouse model by injecting 4T1 cells into the mammary fat pad (Fig. 5a, upper panel indicated by a red arrow). Subsequently, we injected DIR-labeled sEVs into the mouse via the tail vein. As shown in Fig. 5a (lower panel), a signal indicating the presence of DIR-labeled sEVs was detected in the tumor site, suggesting that the tail-vein injected labeled sEVs could accumulate in the tumors. Then, we purified circulating sEVs from HFD-fed mice (H-sEVs) and intravenously injected them into 4T1-bearing CD mice, C-sEVs injection serving as control. The schematic diagram in Fig. 5b illustrates the experimental design. Interestingly, H-sEVs treatment increased the protein levels of ECM1, MMP3, and S100A/B in the tumors when compared to C-sEVs treatments (Fig. 5c). Moreover, H-sEVs treatments significantly increased tumor size (Fig. 5d, e), tumor weight (Fig. 5f), and lung metastasis (Fig. 5g) in the CD mice. These findings suggest that H-sEVs play a role in promoting BC growth and metastasis in vivo.

### ECM1 in the sEVs plays a role in enhancing BC growth and metastasis

To further investigate the role of ECM1 protein in sEVs in promoting BC growth and metastasis, we purified D-sEVs and C-sEVs from C57BL/6 mice. Besides, C-sEVs were loaded with mouse ECM1 construct (ECM-sEVs) in which sEVs served as carriers, allowing the uptake of ECM1 constructs by BC cells and increasing the cellular ECM1 protein levels (Supplementary Fig. S12). These sEVs were then separately injected into CD-fed B6/J-Rab27a-Cas9-KO mice bearing E0771 tumors. The experimental design is illustrated in the schematic diagram in Fig. 6a. Interestingly, we found that treatment with D-sEVs or ECM-sEVs significantly increased the levels of ECM1, MMP3, and S100A/B proteins in the tumors compared to C-sEVs treatments (Fig. 6b). The observed increase in ECM1 levels within the tumor can be attributed to multiple factors, including the enhanced delivery of ECM1 by sEVs to the tumor, increased release of ECM1 from tumor cells, and elevated levels of ECM1 present in sEVs within the tumor microenvironment. Furthermore, mice after receiving D-sEVs or ECM-sEVs had larger tumors (Fig. 6c, d), increased tumor weight (Fig. 6e), and enhanced lung metastasis (Fig. 6f) compared to those receiving C-sEVs.

## Discussion

According to recent statistics from the World Health Organization, there have been 7.8 million reported cases of BC in women over the

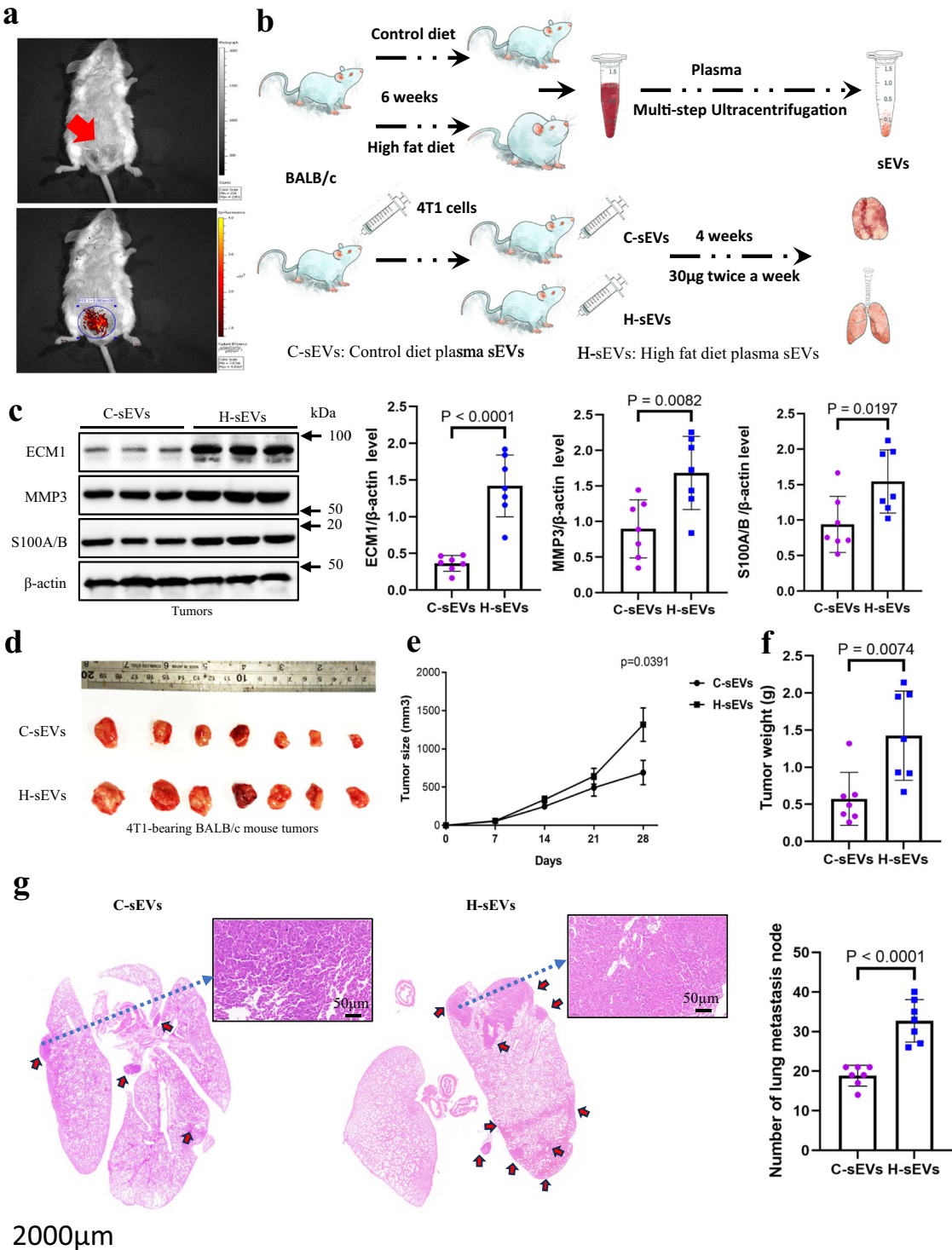

**Fig. 5 | Injection of H-sEVs increases tumor ECM1 levels, cancer growth and metastasis in CD-fed 4T1-bearing mouse model. a** Location of the injected 4T1 cells in the mammary fat pad of the female BALB/c mice (upper panel indicated by an arrow) and accumulation of the tail vein injected DIR-labeled sEVs in the mammary pad of the mice (lower panel). **b** A schematic diagram showing the sEVs treatment protocol. **c** Protein expressions of ECM1, MMP3 and S100A/B in tumors of CD-fed 4T1-bearing BALB/c mice after C-sEVs or H-sEVs treatments. **d** Tumors, **e** tumor size, **f** tumor weight and **g** lung metastasis of the CD-fed 4T1-bearing BALB/c mice after the sEVs treatments. Mouse tumor sizes are presented as the mean ± SEM, other data are presented as mean ± SD; two-sided unpaired *t*-test for (**c**, **f**, **g**); n = 7 mice in each group; *p* values are indicated in graphs. C-sEVs, circulating sEVs in CD-fed BALB/c mice; H-sEVs, circulating sEVs in HFD-fed BALB/c mice; ECM1 extracellular matrix protein 1; MMP3 matrix metallopeptidase 3. Source data are provided as a Source Data file.

past five years, making it the most common cancer worldwide. Numerous studies have demonstrated that obesity is a significant risk factor for BC and negatively impacts BC outcomes, independent of the BC subtypes[35,36].

In our study, we have provided evidence for the significant roles of sEVs and specifically identified ECM1 as a protein in the sEVs that promotes both metastasis and growth of BC under obesity conditions. Additionally, we propose that the elevated levels of ECM1 protein in

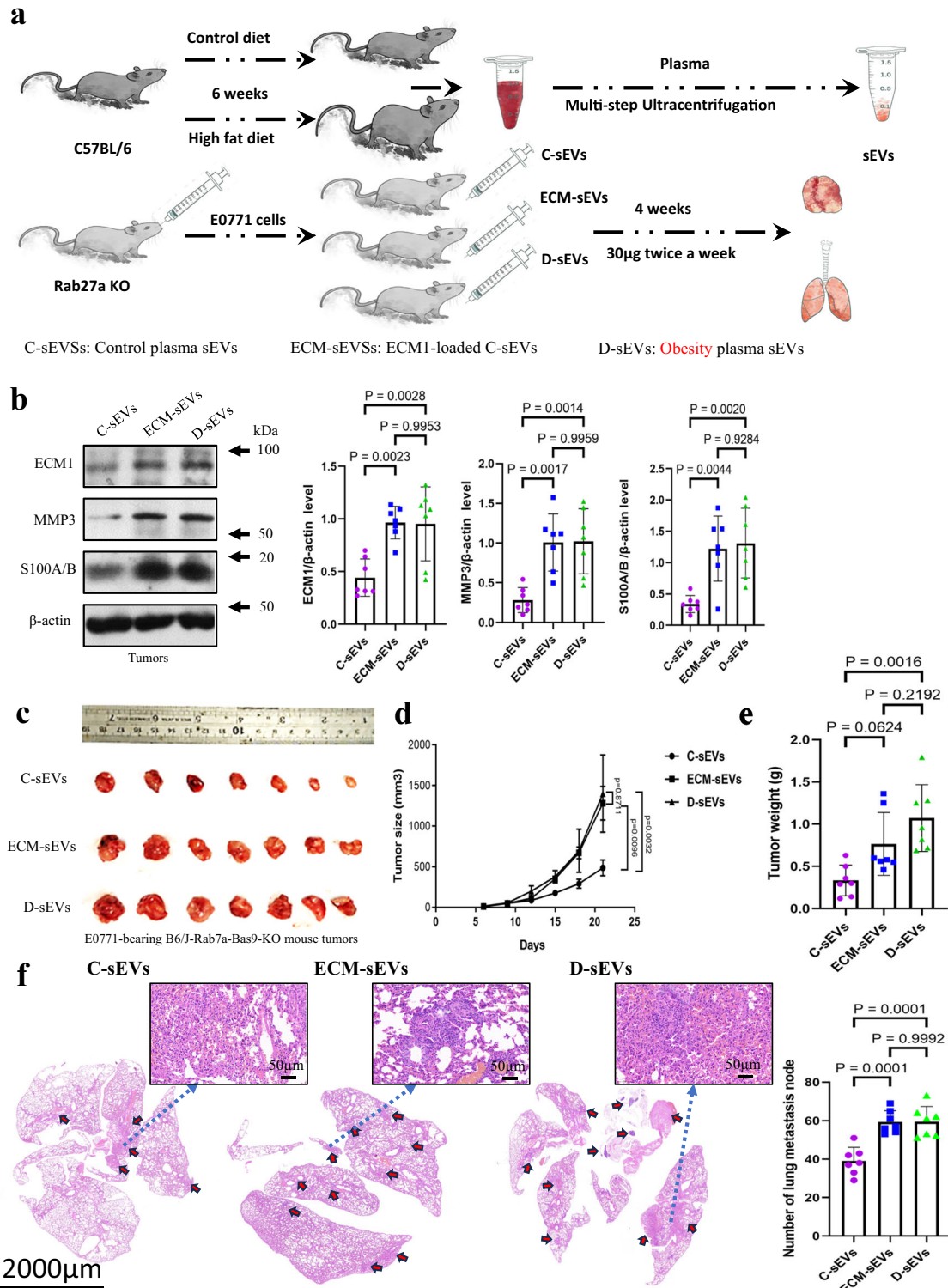

**Fig. 6 | ECM1 in the sEVs plays a role in enhancing BC growth and metastasis. a** A schematic diagram showing the sEVs treatment protocol. **b** Protein expressions of ECM1, MMP3 and S100A/B in the tumors of E0771-bearing B6/J-Rab27a-Cas9-KO mice after C-sEVs, D-sEVs or ECM-sEVs treatments. **c** Tumors, **d** tumor size, **e** tumor weight and **f** lung metastasis of the E0771-bearing B6/J-Rab27a-Cas9-KO mice after sEVs treatments. Mouse tumor sizes are presented as the mean ± SEM, other data are presented as mean ± SD; one-way ANOVA with Tukey's multiple comparison test for (**b, d, e**); n = 7 mice in each group, *p* values are indicated in graphs. Rab27aKO, B6/J-Rab27a-Cas9-KO mice; C-sEVs, circulating sEVs in CD-fed C57BL/6 mice; D-sEVs, circulating sEVs in high-fat-diet induced obesity C57BL/6 mice; ECM-sEVs, ECM1 construct-loaded C-sEVs; ECM1, extracellular matrix protein 1; MMP3, matrix metallopeptidase 3. Source data are provided as a Source Data file.

sEVs are associated with integrin-β2 in the donor cells. The involvement of sEVs in various disease pathologies has been well-documented[24], including their implication in cancers[37], cardiovascular disease[38], neurodegenerative disease[39], and liver disease[40].

By gaining a deeper understanding of the contribution of sEVs to the development of obesity-related BC, we can potentially devise innovative therapeutic strategies that utilize sEVs to improve the current treatment approaches for BC patients with obesity.

ECM1 is a glycosylated protein that plays a crucial role in maintaining the integrity and homeostasis of the skin and connective tissues[41,42]. Clinical studies have demonstrated that ECM1 expressions are higher in BC tissues compared to adjacent epithelial tissues and lymph nodes[43]. Moreover, increased levels of ECM1 in the sera of BC patients have been associated with cancer recurrence and poor long-term survival[44–46].

Interestingly, our study has revealed that the elevated levels of tumor ECM1 protein under obesity conditions are related to the presence of sEVs. Furthermore, our data indicates that the ECM1 protein levels in sEVs are elevated in both mouse models and non-BC individuals with obesity or overweight, suggesting that these sEVs can originate from cell types other than BC. Single-cell sequencing analysis has shown that various cell types, including macrophages and adipocytes, express high levels of ECM1, along with integrin-β2. Integrins serve as transmembrane linkers that facilitate interactions between the cytoskeleton and the extracellular matrix. Stimulation of cells recruits cytoplasmic factors to the cytoplasmic motifs of integrin's beta-chain, leading to integrin activation[47–49]. Recent research has demonstrated that the GPR motif on ECM1 binds to integrin αX and β2, as validated by proximity ligation and co-IP assays using an ECM1 mutant with a VAQ sequence substitution[32]. Interestingly, our findings indicate that while integrin-β2 expression does not affect cellular ECM1 protein levels, it significantly impacts ECM1 protein levels in the sEVs. Knockdown of integrin-β2 reduces ECM1 protein levels in sEVs released by the cells, while overexpression of integrin-β2 increases the ECM1 protein levels. It is suggested that under obesity conditions, elevated integrin-β2 in monocytes and adipocytes enhances ECM1 protein loading into the sEVs. Moreover, the effect of integrin-β2 on the loading of ECM1 protein into sEVs appears to depend on the presence of GRP motif on the protein. The contribution of sEVs derived from adipocytes and macrophages to BC can be influenced by various factors, including abundance of adipocytes and the inflammatory state associated with the number of macrophages. Release of sEVs by these cell types may vary in response to the different impacts of obesity on the pathological conditions. Studies have shown that adipose tissues release a higher quantity of sEVs under obesity[50]. It is plausible to speculate that both macrophages and adipocytes play significant roles in delivering sEVs containing elevated levels of ECM1 to the breast tumors, although the relative contributions of these sEVs cannot be clearly defined.

Treatments of BC cells with D-sEVs lead to a significant increase in the ECM1 protein levels in the released sEVs. In fact, the role of ECM1 in cancer metastasis has been well-documented. In BC, ECM1 was implicated in the control of cancer stem cell-like properties and the process of epithelial-to-mesenchymal transition by stabilizing β-catenin[51]. Notably, ECM1 also contributes to the metastasis of aggressive breast cancer by regulating actin cytoskeletal architecture[28]. Besides, ECM1 induces angiogenesis in BC[52]. Furthermore, ECM1 plays a role in promoting resistance to trastuzumab and inducing the pyruvate kinase M2-mediated Warburg effect by activating epidermal growth factor signaling pathways[53,54]. In our study, we found that the overexpression of ECM1 in BC cells leads to increased expressions of MMP3 and S100A/B. Furthermore, treatment with sEVs purified from DIO or HFD mice not only increases the levels of ECM1 but also enhances MMP3 and S100A/B levels in BC cells in vitro and in tumors in vivo. S100A is a calcium-binding protein of the EF-hand type that is predominantly expressed in cardiomyocytes and skeletal muscle fibers[55]. Although the role of S100A/B in BC is not extensively studied, a report suggests that S100A1 promotes BC growth in xenograft mouse models, and its expression is elevated in BC tissues in humans[56,57]. Our analysis also reveals a significant increase in S100A expression in human BC. MMP3, on the other hand, is a zinc-dependent proteolytic enzyme known for its involvement in angiogenesis, cancer cell growth, and invasion by degrading extracellular matrix and adhesion molecules[58]. Our study is among the few to demonstrate that ECM1 overexpression enhances the expressions of S100A/B and MMP3 in BC. Although the direct or indirect regulation of their expressions by ECM1 in BC is still unknown, the elevated levels of S100A/B and MMP3 may represent a novel mechanism underlying the pro-metastatic role of ECM1 in BC.

Previous research on the molecular link between obesity and BC has primarily focused on adipocyte dysfunction, the release of adipokines such as leptin, and hormonal changes involving insulin and peripheral estrogen aromatization in adipose tissues[10,59,60]. However, our study has shed light on an important role of circulating sEVs in mediating BC growth and metastasis under obesity. It is worth noting that sEVs also contain other constituents, including DNA, RNA, lipids, and metabolites[61]. In addition to revealing the role of ECM1 protein in enhancing BC growth and metastasis under obesity, it would be intriguing to investigate other cargo contents within the sEVs that may also contribute to BC development.

Despite the negative impact of obesity on BC outcomes, the efficacy of current systemic chemotherapy and endocrine therapy in BC patients with obesity is limited[62]. Clinical trials such as ATAC (Arimidex, Tamoxifen, Alone or in Combination) have shown that the relative benefits of these treatments in terms of total and distant recurrences are absent or diminished in BC patients with obesity compared to normal weight patients[63]. Similar findings have been reported in several other clinical studies[64,65]. Moreover, recent research has indicated that these BC patients exhibit reduced sensitivity to anti-vascular endothelial growth factor-based therapy due to elevated systemic levels of interleukin-6 and fibroblast growth factor-2[66]. The findings from our study, highlighting the role of circulating sEVs in promoting BC growth and metastasis under obesity conditions, which may suggest therapeutic strategies centered around sEV-based interventions for the treatment of obesity-associated BC.

In summary, our study is to highlight the significant role of ECM1 protein present in circulating sEVs, which contributes to the increased growth and metastasis of BC under obesity conditions. This novel mechanism elucidated in our study not only provides valuable insights but also proposes a potential sEV-based therapeutic approach for treating obesity-associated BC.

## Methods

### Human 4D-label free quantitative proteomics

This clinical study was approved by the Ethics Committee at Hangzhou Xixi Hospital (Zhejiang province, China). We obtained informed consent form of all the participants without compensation. A total of 48 healthy people (11 males and 37 females, $34.19 \pm 11.98$ years old) with BMI $\leq 23$ and without acute and chronic diseases were included in the normal weight healthy control group; A total of 96 people (77 males and 19 females, $37.19 \pm 13.53$ years old) with BMI $\geq 25$ and without any infectious diseases, cancer, surgery, and genetic diseases were included in the group with obesity (Details in Supplementary Data, Human subject information). A total of 0.5 mL of the EDTA anticoagulant plasma was taken from each participant in the clinical tests. Plasma of 8 participants were combined into one sample tube with a total of 4 mL in volume. A sample tube in each group was used for sEVs extraction for subsequent studies, the remaining sample tubes were used for proteomics study. The female analysis cannot be carried out because of the gender-mixed blood.

Proteomic analysis was done at Shanghai Hwayen Biotechnological, Inc., China. The vacuum-dried samples were reconstituted with 0.1% formic acid. EASY-nLC 1200 (ThermoScientific), analytical column (C18, 1.9 um, 75 μm x 20 cm) with a flow rate of 200 nL/min were used for separation. The mass spectrometer was an Orbitrap Exploris 480 + FAIMS Pro (ThermoScientific). Data Dependent Acquisition (DDA) mode was used for tandem mass spectrometry detection. The full scan resolution was 60,000 (FWHM). The mass-to-charge ratio range was set to m/z 350–1600, and in HCD fragmentation mode, the collision energy was set to 30%. Database Proteome Discoverer 2.4 was

used as a proteomic analysis software platform. It was specially designed for high-resolution MS data analysis, and it provides comprehensive proteomic qualitative determination volume data viewing. The protein profiles from different samples might be different, in the label free quantitative proteome detection data, the identified proteins did not exist uniformly in all the samples. The protein deletion value of a single sample was set at 10% to 30%. In the process of data analysis, the missing values were handled based on the following criteria: firstly, the protein with missing quantitative value was removed; secondly, the missing value of the protein that was detected in more than one sample would be adjusted to half of the minimum value of the quantitative value. Quantitative statistical analysis was done after data filling. Continuous variable statistical analysis was performed for the comparison between the two groups. All the proteins were analyzed according to the groups by Student's t-test or Mann Whitney U-test that calculated the consistencies of all the proteins in the groups. Frequency statistical analysis was performed for proteins that were not detected in all the samples but have quantitative values in one or more than one samples, Chi-square test and Fisher's exact test were performed. If the detection rate of proteins in the group was ≥ 60%, the significant difference between groups was obtained by continuous variable statistical method. The results were used for subsequent biological information analysis and data mining. The proteomics data has been deposited to Pride (ProteomeXchange). Please find IDs in Data Availability section.

## Multiple reaction monitoring (MRM)-mass spectrometry

The expressions of the human differential expressed proteins (DEPs) in the sEVs were validated by MRM. MRM is a mass spectrometry analysis of target molecules technology using triple quadrupole mass spectrometer to detect the mass spectral response signals of the parent ions and product ions of the target molecule. MRM obtains sensitive and reproducible qualitative and quantitative information. Experimental settings of MRM were as follows: For the liquid phase conditions: Phase A was 2% ACN, 0.1% FA. Phase B was 98% ACN, 0.1% FA. The flow rate was 5ul/min. Gradient conditions were 0–5 min, 95%A, 5%B; 5 ~ 105 min, 70%, 30%B; 105 ~ 115 min, 20%A, 80%B; 115 ~ 120 min, 98%A, 2%B. For the mass spectrometry conditions: The ion source: electrospray ion source (ESI); positive ion mode detection; Scanning mode: multiple reaction monitoring (MRM); injection voltage: 5500 eV; temperature: 150 °C; curtain gas (CUR, N2): 30 psi; collision gas pressure (CAD, N2): High mode, the auxiliary gas GAS1 pressure=20 psi; The auxiliary gas GAS2 pressure: 15 psi; The scan time: 10 ms. Degrouping voltage (DP) and collision energy (CE) were detailed in Supplementary Data 5. The scheduled MRM acquisition method was used, the total scan time was 1.7 S, and the MRM detection window was 300 S.

A total of 30 selected protein expressions were examined by MRM; the information of the proteins and peptide sequences are listed in Supplementary Data 3. The MRM raw data of all samples were imported into Skyline software for quantitative statistics. To improve the quantitative statistical accuracy and signal-to-noise ratio (S/N), statistical correction is performed according to the following principles: a) Select at least one peptide to calculate the quantitative result of a target protein; b) Select at least three transitions to calculate the quantitative results of the target peptide; c) Manually remove any overlapping transitions with saturated signal-to-noise ratio <5. The quantitative result of MRM statistics was based on the peak area (area) of the chromatographic peak of the ion. The quantitative value of the target protein was calculated by weighting the total area of several polypeptides of the protein. The total area of the target peptide was calculated by weighting the fragment ion area of the peptide.

## Mice

Animal experiments were conducted in accordance with the guidelines and with the approval of the Committees of Animal Ethics and Experimental Safety of Hong Kong Baptist University and procedures were approved by the Department of Health under Hong Kong legislation. All C57BL/6 female mice and BALB/c female mice of 6 to 8-week-old were purchased from the Laboratory Animal Services Centre of The Chinese University of Hong Kong. All mice were bred and housed in the animal facilities at School of Chinese Medicine at The Hong Kong Baptist University on a 12 h light/dark cycle with constant ambient temperature (22–24 °C) and humidity (~60%). They were fed with either standard laboratory chow (Labdiet 5053), high fat diet (HFD) (Research Diets D12492, 60 kcal% fat) or matched control diet (CD) (D12450J, 10% kcal% fat) and applied with water *ad libitum*. The animal procedures were suggested in Figs. 5b and 6a (all n = 7 each group).

The euthanasia of the mice was decided by cervical dislocation when the following situations was occurred.

a) The animal is on the verge of death or cannot move, or does not respond after being given gentle stimulation; b) Dyspnea: Typical symptoms are salivation and/or cyanosis in the mouth and nose; c) Incontinence of urine and feces; d) Body weight reduced by 20% of pre-experiment body weight; e) Inability to eat or drink; f) The animal shows obvious anxiety, restlessness; g) The weight of the tumor exceeds 10% of the animal's own body weight; h) Paralysis, persistent epilepsy or stereotyped behavior, etc.; i) The animal's skin damage area accounts for more than 30% of the whole body, or infection and suppuration occur; j) Other situations determined by a veterinarian to require a humane endpoint.

## Establish constitutive Rab27a knockout mice (B6/J-Rab27a-Cas9-KO)

Constitutive Rab27a knockout mice (B6/J-Rab27a-Cas9-KO) were obtained by CRISPR/Cas9 technology (GemPharmatech). CRISPR-Cas9 system was used to perform the genome editing in the mice by delivering guide RNA to the designated genomic site and knockout Rab27a expression. Knockout of Rab27a in the mouse was confirmed by genotyping. The primers sequences for genotyping were JS05042-Rab27a-5wt-t, forward (F1) 5'- AGGCTACACGTACTGTTTCAAGGG-3' and reverse (R1) 5'- AAACCAACAGTGTCAGCCATGTGTC-3', and JS15042-Rab27a-wt-t, forward (F2) 5'- AGCTTTGTGGATGCTAAGCAAGAC-3' and reverse (R2) 5'- TCATCTTACCAGGTGCAGATGAG-3'. Western blot analysis was also used to validate the knockout of Rab27a protein in various adipose tissues and organs. Rab27a knockout female mice (n = 35) of 6 to 8-week-old were used for experiments.

## Establish high-fat diet or high-fat deit-induced obesity (DIO) mouse model

BALB/c female mice or C57BL/6 of 6 to 8-week-old (n = 150, respectively) were fed with either a HFD (Research Diets D12492, 60 kcal% fat) for 6 weeks to establish HFD-induced obesity (DIO) mouse model or a matched control diet (Research Diets D12450J, 10% kcal% fat) served as controls. Mouse body weight was measured twice a week. These mice were sacrificed for the subsequent experiments.

## Establish metastatic breast cancer mouse model

Before cancer cells inoculation, the mice were fed with either HFD (Research Diets D12492, 60 kcal% fat) or CD (Research Diets D12450J, 10% kcal% fat) for 6 weeks. The body weight and tumor size of mice were measured every 3 days. BALB/c female mice of 6 to 8-week-old (n = 14) were injected $10^6$ x 4T1 cells in Matrigel suspension (1:1:0.2 mL) into the fourth mammary fat pad. C57BL/6 mice (n = 14) were injected $6 \times 10^5$ E0771 cells in Matrigel suspension (0.2 mL) into the fourth mammary fat pad. Rab27aKO mice of 6 to 8-week-old (n = 35) were injected $6 \times 10^5$ E0771 cells in Matrigel suspension (0.2 mL) into the fourth mammary fat pad. Mice were sacrificed, plasma, tumors and lungs were dissected for subsequent experiments. The maximal tumor burden permitted was 5% of the animal's body weight under the guideline of the Committees of Animal Ethics and Experimental Safety

of Hong Kong Baptist University. The all-tumors sizes in this study were not exceeded the permitted maximal tumor burden.

## Purification of sEVs from human and mouse plasma

Differential centrifugations were used to isolate the sEVs in the plasma. Human plasma (4 mL) was diluted with 30 mL PBS, and mouse plasma (0.5 mL) were diluted with 1 mL PBS. Then, the diluted plasma was centrifuged to get supernatant at 2,000 × g for 10 min, and 10,000 × g for 30 min at 4 °C to eliminate dead cells and cell debris. Then they were centrifugated at 160,000 × g for 1 hr (Beckman Coulter) twice to get the pellets that were resuspended in PBS followed by filtration with 0.22 μm membrane filters. Finally, the pellets resuspensions were centrifuged at 160,000 × g for 1 hr to get the sEVs.

## Depletion of sEVs from plasma sample

Mouse plasma (1.9 mL) was centrifuged to get supernatant at 2,000 × g for 10 min, and 10,000 × g for 30 min at 4 °C to remove dead cells and cell debris. Then they were centrifugated at 160,000 × g for 15 hr (Beckman Coulter), and only supernatant was collected. CD63 expression was used to indicate the removal of sEVs from the samples.

## Purification of sEVs from conditioned medium

Total sEVs were purified from the cell culture media with Total Exosome Isolation kit (Invitrogen). Firstly, cell culture media was centrifuged at 2,000 × g for 30 min to remove cells and debris. The supernatant containing the cell-free culture media was transferred to a new tube without disturbing the pellet. Total sEVs isolation reagent (1/2 volume) was added into the supernatant and incubated overnight at 4 °C. After incubation, samples were centrifuged at 10,000 × g for 1 hr at 4 °C. The pellet (sEVs) was resuspended in PBS for subsequent studies.

## Purification of sEVs released from subcutaneous adipose tissues

Subcutaneous adipose tissues (SAT) were dissected from mice, the adipose tissues were cut into small pieces and cultured in the DMEM medium prepared by sEV-depleted FBS (ThermoFisher Scientific) for 24 hr. The sEVs released from SAT were extracted as described above.

## Authentication of the purified sEVs

Human and mouse plasma sEVs were visualized by negative stained transmission electron microscopy (TEM) at Shanghai Hwayen Biotechnological, Inc., China. Human and mouse plasma sEVs particle sizes were analyzed by nanoparticle tracking analysis (NTA, PARTICLE METRIX, ZetaVIEW S/N 22–756) at Shanghai Hwayen Biotechnological, Inc., China, and the protein expressions of human sEVs markers including CD63, CD9, TST10 and calnexin were examined. Expressions of mouse sEVs markers including CD63 and CD81 were examined by Western blot.

## Mouse iTRAQ-based quantitative proteomics

This animal study was approved by the Ethics Committee at the Hong Kong Baptist University. A total of 4 female and 4 male C57BL/6 mice were fed HFD (Research Diets D12492, 60 kcal% fat) for 6 weeks (BT). A total of 4 female and 4 male C57BL/6 mice were fed matched control diet (Research Diets D12450J, 10% kcal fat) for 6 weeks (BC). At the end of the dietary intervention, the body weight of BT group was significantly higher than BC group. Multiple ultracentrifugation was used to purify sEVs from mouse plasma, the proteomic analysis was done at Guangzhou Fitgene Biotech Co., LTD, China.

Protein samples were enzymatically hydrolyzed by trypsin. After enzymatic hydrolysis, the protein was cleaved into small peptides for subsequent mass spectrometry analysis. Peptides in different samples were labeled separately with ITRAQ reagent. ITRAQ reagent is a chemical reagent containing isotope markers that reacts with amino groups in peptides to form labeled peptides. The labeled peptides were mixed to form a composite sample. Peptides in complex samples were separated by liquid chromatography (first dimension: LC-20AD, Shimadzu; second dimension: Dionex Ultimate 3000 RSLCnano, Thermo Scientific). The separated peptides were analyzed on Q Exactive Orbitrap Mass Spectrometers (MS, Thermo Scientific). The first-level mass spectrometry parameters were: Resolution: 70,000; AGC target: 3e6; Maximum injection time: 100 ms; Scan range: 350 to 1800 m/z. The secondary mass spectrometry parameters were: Resolution: 17,500; AGC target: 5e4; Maximum injection time: 120 ms; Top precursor ions: 20; Stepped Normalized Collision Energy (SNCE): 30. The basic settings of data analysis were: Detected Protein Threshold [Unused ProtScore (Conf)] > 0.05(10%); Competitor Error Margin (ProtScore): 2; Paragon™ Algorithm: 4.5.0.0,1654; Annotations Retrieved from UniProt; Sample Type: iTRAQ 8plex (Peptide Labeled); Cys. Alkylation: MMTS; Instrument: Orbi MS (1–3ppm), Orbi MS/MS.

With mass spectrometry analysis, the mass spectral peak of the peptide was obtained, and its intensity was measured. The mass spectrometer acquired raw mass spectrometry data in RAW format, and then used Proteome Discoverer 1.4 (Version 1.4.0.288, Thermo Fisher) to convert the RAW file to an MGF format mass spectrometry file and entered the MGF format mass spectrometry file and protein retrieval library into ProteinPilot™ Software 4.5 (version 1656, AB Sciex) for mass spectrometry search. The original data retrieved by the ProteinPilot software was analyzed, and the analysis process: 1) After the retrieval was completed, filter the Unused value for the original search results, and set Unused≥1.3 to make the protein trustworthy. The degree is above 95%; 2) To remove the anti-database in the search results, and to remove the records starting with "RRRRR" in the search results. ProteinPilot™ was performed FDR analysis. The protein data obtained by the above step analysis was a trusted protein. Differential screening for trusted proteins, screening process:1) remove proteins without quantitative information; 2) Number of peptides≥ 2; 3) The mean value of the data ratio of each group was calculated, and BT: BC ≥ 1.3 was selected as the upregulated protein and BT: BC ≤ 0.77 was the downregulated protein; The data obtained by the above step analysis were differentially expressed proteins. The proteomics data has been deposited to Pride (ProteomeXchange). Please find IDs in Data Availability section.

## PKH67 and DIR staining and ECM1 immunofluorescence labeling

sEVs purified from the C57BL/6 mouse plasma were suspended in 100 μL PBS before incubating with 0.5 μL PKH67 (Sigma-Aldrich) at room temperature for 30 min in dark. The labeled sEVs were resuspended in 300 μL 1% BSA-containing PBS before 1.9 mL PBS was added and ultra-centrifugated at 160,000 × g for 1 hr. Pellets were washed with PBS and ultra-centrifuged at 160,000 × g for 1 hr. The PKH67-labeled sEVs were used to treat the 4T1 cells or E0771 in 24-well plate overnight. After overnight treatments, the cells were co-cultured with 1 μL ECM1 antibody for 1 hr, and then conjugated to 1 μL Alexa Fluor® 594 1 hr for fluorescent imaging. The cells were also stained with 0.5 μL DIR dye (ThermoFishier Scientific) for 30 min in dark. Then the cells were fixed by methanol for 10 min, and stained by 0.1 μg/mL 4',6-diamidino-2-phenylindole (DAPI) for 5 min. The signals were detected by Confocal Laser Scanning Microscopy (Leica).

## In vivo imaging of labeled sEVs

Plasma sEVs (30 μg) purified from 4T1-bearing BALB/c mice were suspended in 100 μL PBS before mixing with 0.5 μL DIR dye (Thermo-Fishier Scientific) at room temperature for 30 min in dark. The labeled sEVs were mixed with 300 μL of 1% BSA-containing PBS for 1 min, before PBS of 1.9 mL was added followed by ultracentrifuge at 160,000 × g for 1 hr. Pellets were washed with PBS and ultracentrifuged at 160,000 × g for 1 hr. Then, the pellets were resuspended in 100 μL PBS, and were injected into the tail vein of the 4T1-bearing BALB/c mice after anesthesia. Images were acquired with IVIS Lumina XR system under ICG (810–875 nm wavelength).

## Loading ECM1-construct into sEVs

Plasma sEVs from control diet (CD) mice (C-sEVs) were loaded with mouse ECM1 construct (SinoBiological MG50331-ACG) using Exo-FectTM exosome transfection reagent (System Biosciences). Briefly, the plasma sEVs were purified from CD-fed C57BL/6 mice and the sEVs protein concentration was measured by BCA assay. In a 1.5 ml tube, we mixed the followings with volume 150 μL in total: 10 μL Exo-Fect solution, 20 μL plasmid DNA (5 μg), 70 μL sterile 1x PBS, 50 μL purified sEVs (50 μg). The mixture was incubated at 37 °C on a shaker for 10 min. The reaction was stopped by mixing with 30 μL ExoQuick-TC reagent on ice for 30 min. The ECM1 construct-loaded sEVs were pelleted by centrifugation at 18407 × g for 3 min.

## sEVs treatments

For the in vitro treatments, 4T1 (5 ×10$^4$ cells) or E0771 (5 ×10$^4$ cells) were cultured in RPMI 1640 or DMEM full medium with HEPES in 24-well plate overnight. Then, the full medium was replaced by the culture medium prepared by sEVs-depleted FBS (ThermoFisher Scientific). sEVs of 30 μg/mL were used to treat the cells for 18 or 21 hr. To explore whether the sEVs-treated BC cells would release sEVs with enhanced ECM1 protein levels, after treating the cells with the plasma sEVs (30 μg) in culture medium prepared by sEVs-depleted FBS for 21 hr, we changed the culture medium with fresh medium prepared by sEVs-depleted FBS and cultured the cells for another 24 hr. Then we collected the sEVs released from these cells as described above. ECM1 protein levels in the sEVs were examined by Western blot.

For in vivo treatments, sEVs (30 μg) were resuspended in PBS (100 μL) and were injected into the tail vein of 4T1-bearing BALB/c, E0771-bearing C57BL/6 or E0771-bearing Rab27aKO female mice (n = 7 each group) twice a week. The sEVs treatments lasted for 3 weeks for C57BL/6 and Rab27aKO female mice, and 4 weeks for BALB/c female mice. The body weight and tumor size of mice were measured every 3 days. The mice were sacrificed, plasma, tumors and lungs were dissected for subsequent experiments.

## Hematoxylin and eosin (H&E) staining

H&E staining was used for primary diagnosis of breast cancer lung metastasis. The fresh lung tissue was dissected and fixed in the pre-prepared 10% formalin fixative solution, then orderly conducted the procedures of dehydrated transparent, soaked wax embedding, slicing with patches, dewaxing, staining, dehydrated transparent, and sealing. Finally, the sealed slides were observed under light microscopy (Leica).

## Cell culture

4T1, E0771, HCC1806, and RAW 264.7 cells were purchased from the American Type Culture Collection (ATCC, USA; CRL-2539 for 4T1, CRL-3461 for E0771, CRL-2335 for HCC1806, TIB-71 for RAW 264.7). HCC 1806 and 4T1 cells were cultured in RPMI 1640 medium with 10% FBS, 1% penicillin and 1% streptomycin (Invitrogen). E0771 and RAW 264.7 cells were cultured in DMEM medium with 10% FBS, 1% penicillin, 1% streptomycin, and 20 mM HEPES. Cell passage was done every 3–5 days. The cells were cultured at 37 °C in a humidified incubator with 5% CO$_2$.

## siRNA transfection

After loading siRNA against integrin subunit beta-2 (integrin-β2) (Shanghai GenePharm) (5′-GUCCAGCCGAUGAUAUCAAUU-3′, 5′-AUGAUAUCAUCGGCUGGACUU-3′) or the negative control (5′-UUCUCCGAACGUGUCACGUTT-3′, 5′-ACGUGACACGUUCGGAGAATT-3′) to RAW264.7 cells (1×10$^5$) in 12-well plate by Lipofectamine RNAimax (Invitrogen) for 72 hr, the full medium was changed to culture medium prepared by sEVs-depleted FBS (ThermoFisher Scientific) overnight before sEVs extraction. ITGB2 mRNA expression in RAW264.7 cells was examined by real time PCR. Cellular integrin-β2 and ECM1, and sEVs integrin-β2, ECM1, ICAM1, and GPX3 expressions were examined by Western blot.

## Overexpression of integrin-β2 in RAW264.7 cells

RAW264.7 cells (1 × 10$^5$ cells) were transfected with human integrin-β2 construct (2 μg) (SinoBiological MG 50359-UT) or the empty vector (2 μg) using Lipofectamine 3000 (Invitrogen) following manufacturer's instructions. Six-hour post transfection, cells were cultured in full medium for another 48 hr. Twenty-four hr before sEVs extraction, the full medium was changed to culture medium prepared by sEVs-depleted FBS (ThermoFisher Scientific).

## Overexpression of ECM1 in BC cells

HCC1806 cells were transfected with human ECM1 construct (SinoBiological HT10362-UT) using Lipofectamine 3000 (Invitrogen) following manufacturer's instructions. Briefly, 1 × 10$^5$ HCC1806 cells were plated onto 12 well plates overnight. ECM1 construct (2 μg) or pcDNA3.1 empty vector (2 μg) was used for transfection. Six-hour post transfection, cells were cultured in full medium for another 48 hr.

## Real-time qPCR

Total RNA from cells was extracted with TRIzol Reagent (Invitrogen) and reverse transcribed with PrimeScript RT reagent Kit (TaKaRa) following manufacturer's instructions. Real-time PCR was performed using SYBR Premix Ex Taq II Kit (Takara) by ViiA7 applied biosystems (Life technologies). β-actin was used as a normalization control. The primer sequences for mouse *ECM1* gene were forward: 5′-CGACACAAACCGCCTAGACTGT-3′ and reverse 5′-CTGGAAGCAAGAGAATCGCTCC-3′; mouse *ITGB2* gene were forward 5′-ATGTGGGCCCACACTCACTGC-3′ and reverse 5′-TTAACAAAAGGCAGCACCGT-3; mouse β-actin gene were forward: 5′-CATTGCTGACAGGATGCAGAAGG 3′ and reverse: 5′ -TGCTGGAAGGTGGACAGTGAGG-3′.

## Western blot analysis

Total protein of the cells was extracted using RIPA buffer on ice. Organs or tumors were ground with RIPA buffer in a tissue homogenizer at room temperature for 45 sec, repeated three times. After centrifugation at 10,000 × g for 15 min, the lysed cells or remaining tissues were removed. The protein concentration was determined using the BCA assay. An equal amount of protein was then loaded onto SDS-PAGE and transferred onto polyvinylidene fluoride membranes. Following blocking with 5% milk, the membranes were incubated overnight with the corresponding primary antibody. Horseradish peroxidase-labeled secondary antibody was subsequently added, and visualization was achieved using Western HRP substrate (EMD Millipore). The following antibodies were used in this study, ECM1 (1:1000, Santa Cruz, sc365335), integrin β2 (1:1000, Cell Signaling Technology, #47985) MMP3 (1:600, Santa Cruz, sc-21732), S100A/B (1:1000, Novus Biologicals, NBP2-29403), GPX3 (1:500, Abcam, ab256470), ICAM-1(1:600, Santa Cruz, sc8439), β-actin (1:5000, Santa Cruz, sc81178), Rab27a (1:1000, Cell Signaling Technology, CST69295S), CD63 (1:500, Abcam, ab216130), CD81(1:1000, Santa Cruz, sc166029), CD9 (1:1000, SBI, EXOAB-CD9A-1), TSG10 (1:1000, SBI, EXOAB-TSG101-1), CD63 (1:1000, SBI, EXOAB-CD63A-1), CD81 (1:1000, SBI, EXOAB-CD81A-1), Calnexin (1:500, Immunoway, YT0613). The protein band intensities were assessed by Image J software version 1.48. Full scan blots are provided in Source Data files.

## Breast cancer cell migration and invasion

For the cancer cell migration experiments, HCC1806, 4T1, or E0771 cells were harvested and suspended in either serum-free RPMI 1640 or DMEM medium. A 250 μL cell suspension (2 × 10$^5$ cells/mL) with sEVs (30 μg/mL) or E0771-derived extracellular matrix (50 μg) were seeded into the top chambers of transparent PET membranes (8-μm pore size, Falcon). The cells were allowed to migrate for 18–24 hr. In the bottom chambers, culture medium (RPMI 1640 or DMEM) containing 15% FBS was added.

In the cell invasion experiments, HCC1806, 4T1, or E0771 cells were harvested and suspended in either serum-free RPMI 1640 or DMEM medium. A 250 μL cell suspension ($4 \times 10^5$ cells/mL) with sEVs (30 μg/mL) or E0771-derived extracellular matrix (50 μg) were seeded into the top chambers with PET membranes (8-μm pore size) that were pre-coated with Matrigel (25 mg/mL). The bottom chamber was filled with culture medium (RPMI 1640 or DMEM) containing 15% FBS. The cells were incubated for 24 hr. After incubation, the cells on the upper surface of the membrane were removed using cotton swabs. The cells on the lower surface of the membrane were fixed with methanol for 5 min and stained with 1% crystal violet solution for 5 min. Cell numbers were counted in randomly selected fields under a light microscope at 400× magnification, or OD values were obtained at a wavelength of 750 nm.

## ELISA for ECM1

E0771 ($1 \times 10^5$) cells was treated with either C-sEVs or D-sEVs (30 μg/mL) in culture medium prepared by sEVs-depleted FBS (ThermoFisher Scientific) for 24 hr. To isolate the cell-derived extracellular matrix, TritonX-100 (1%) was added to the E0771 cells and then washed three times with sterile ultra-pure water before centrifugation at 503 × g for 10 min[67]. The cell-derived extracellular matrix was obtained by scraping the cells in 100 μL PBS. ECM1 levels in the E0771-derived extracellular matrix samples were measured by Mouse ECM1 ELISA kit (EK Biosciences). Each sample (50 μg) was incubated with 100 μL ECM1 antibody-HRP for 60 minutes at 37 °C. After washing five times, substrates A and B (50 μL each) were added and incubated at 37 °C in the dark for 15 min. Stop solution (50 μL) was added to the sample. OD values were measured at 450 nm wavelength.

## Immunoprecipitation of ECM1

E0771-derived extracellular matrix samples were used for immunoprecipitate ECM1 protein. Protein G MagBeads (25 μL, Genescript C42742207A) was placed on a magnetic separator so that the immunomagnetic beads could be separated from the solution and the supernatant was removed. ECM1 antibody (2 μL, Santa Cruz, sc365335) was diluted in the 200 μL PBST with beads, which was incubated and spined down at room temperature for 10 min. After placing the ECM1 antibody-beads complex on a magnetic separator and removing the supernatant, the ECM1 antibody-bead complex was washed by 200 μL PBST. The sample (100 μL) was incubated with the ECM1 antibody-beads complex by spinning at room temperature for 30 min. Then it was placed on a magnetic separator and the supernatant was transferred to a clean tube for further analysis. The removal of ECM1 from the samples was validated by ELISA.

## Virtual flow analysis and prognosis analysis

RNA-sequencing expression (level 3) profiles and corresponding clinical information for breast cancer were downloaded from the TCGA dataset (https://portal.gdc.com). A total of 898 breast cancer patients were divided into 4 stages, including G 1(N = 333), G 2 (N = 366), G 3 (N = 120), G 4 (N = 79). A total of 572 normal tissues (N = 113 para-cancer tissue from TCGA database; N = 459 from Genotype-Tissue Expression database) were included as controls. R software GSVA package was used to analyze the correlation between ECM1 and pathway scores by Spearman correlation, choosing parameter as method = 'ssgsea'. R software ggplot2 package was used to analyze the expression distribution of ECM1 gene in tumor tissues and normal tissues. All the analysis methods and R packages were implemented by R version 4.0.3.

## Single cells RNA analysis of ECM1 and ITGB2

The ECM1 and ITGB2 RNA expressions for single cell RNA sequence were obtained from the database of the human protein atlas: https://www.proteinatlas.org. ECM1 and ITGB2 RNA expressions in different cell types were analyzed. ECM1 RNA was enriched in mesenchymal cells, trophoblast cells, blood and immune cells. In blood and immune cells, the ECM1 RNA was enriched in macrophages. ITGB2 RNA was enriched in blood and immune cells, including macrophages.

## Statistics and reproducibility

Each experiment was independently performed at least three times. Animal experiments involved at least three independent and randomly chosen mice at comparable developmental stages and none of the samples were excluded from analyses. Sample size was determined from the power of the statistical test performed and was increased in accordance with the statistical variation. Mouse tumor sizes were presented as the mean ± SEM, other experimental data were presented as mean ± SD. The two-sided unpaired *t*-test was used for statistical evaluation for two-group comparisons. Ordinary one-way ANOVA with Tukey's multiple comparison test for three-group comparisons. The Kruskal-Wallis test with the Wilcoxon's multiple comparison for four groups comparisons. All statistical analyses were performed with Prism software (GraphPad Prism for Windows, version 9.0). All data meet the normal distribution. P value < 0.05 was considered statistically significant.

## Reporting summary

Further information on research design is available in the Nature Portfolio Reporting Summary linked to this article.

## Data availability

All relevant data supporting the key findings of this study are available within the article and its Supplementary information files. The datasets analyzed during the current study are available in PRIDE under accession code PXD041236 and PXD041294 for sEVs proteomics, Human Protein Atlas (https://www.proteinatlas.org; January 10, 2023) for single-cell RNA-seq profile of ECM1 and ITGB2, The Cancer Genome Atlas Program (https://www.cancer.gov/ccg/research/genome-sequencing/tcga; January 26, 2023) for RNA-seq profiles of breast cancer bulk, and Genotype-Tissue Expression database (https://gtexportal.org/home/; January 26, 2023) for bulk RNA-seq profiles of normal breast tissue. Source data are provided with this paper.

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

## Acknowledgements

We thanked the Electron Microscope Unit at the Queen Mary Hospital at the University of Hong Kong for providing the TEM services; the Shanghai Hwayen Biotechnological Inc for providing the human proteomics and MRM-mass spectrometry services; the Guangzhou Fitgene Biotechnological Inc for providing the mice proteomics; Liu Lijuan and Wang Wuyue for designing the elements for Fig. 5b and 6a. This work was partially supported by HMRF (08193596), FNRA-IG (RC-FNRA-IG/20-21/SCM/01), Shenzhen Basic Research Program for Shenzhen Virtual University Park (2021Szvup131), GDNSF (2021A1515010655 and 2023A1515011811) and ITC (PRP/015/19FX) to HY Kwan; Project of Hangzhou Science and Technology Bureau (20201203B179, 2021WJCY061, 2021WJCY186) and Project of Zhejiang Provincial Department of Health (2022KY1019) to JF Bao.

## Author contributions

K.X., C.L., T.S., T.T., J.B., A.L., H.Y.K. contributed conception and design of the study. K.X., A.F., Z.L., L.M., Z.L., M.C., K.J. performed the experiments and interpreted the data. K.X. and H.Y.K. wrote the manuscript. C.L., T.S., T.T., J.B., A.L., H.Y.K. revised the manuscript. All authors have given approval to the final version of the manuscript.

## Competing interests

The authors declare no competing interests.
