## [Peer Review File · Nature Communications]

REVIEWER COMMENTS

Reviewer #1 (Remarks to the Author): with expertise in breast cancer, obesity

Circulating exosomal ECM1 protein level is elevated in association with integrin- β 2 and mediates the enhanced breast cancer growth and metastasis under obesity conditions by Xu et al.

This work by Xu et al described circulating exosomal ECM1 is elevated in obesity, contributing to breast cancer growth and metastasis. There is a correlation between exosomal ECM1 and integrin- β 2 co-loading into the exosomes. The origin of exosomes can be from monocytes/macrophages or adipocytes induced under obese conditions. Forced loading of ECM1 in exosomes can promote tumor growth and metastasis of breast cancer models. This project is interesting and has some potential impact to promote mechanistic understanding of how obesity promotes breast cancer progression and metastasis. Though this work has intrinsic innovation at the conceptual level, this manuscript, however, has some major issues related to data quality, in depth mechanisms and the significance in breast cancer relevance.

Major concerns:

- 1) All data quality is very low and the legends in the pictures are illegible, examples including all bar graphs, images, and many other labels.
- 2) The major animal experiments are all with 3 mice per group; it is way under powered. For any tumor studies, even with 1.5 variations in volume between groups and with 0.8 power, a minimal of 7-8 mice per group should be required. For example, images shown in Figure 6C seem to have much bigger than 1.5 variations and the statistics in figure 6d did not recapitulate such variations.
- 3) The pathological connection between ECM1 and breast cancer metastasis is established by using whole tumor RNAseq dataset, with most ECM1 transcripts expected to be expressed by cancer cells. Cancer cells overexpressing ECM1 also exhibit more tumor growth and metastasis. How much ECM1 can be achieved under physiological conditions to induce breast cancer progression and metastasis? ECM1 is expected a secreted protein working with cancer cells within the tumor microenvironment, it is logic to presume that most microenvironmental ECM1 comes from cancer cells or stromal cells, other than from exosomes. Figure S7b showed HFD includes ECM1 expression in tumors, but how can the authors prove those are secreted from tumors into the microenvironment or those come from serum exosomes?

4) The study lacks of mechanistic explanation for exosomal ECM1 to work in cancer cells. ECM1 is a secreted protein and mostly known for its function. Cancer cells pick exosomes and take the exosomal proteins within cells. Does ECM1 from exosome need to be secreted by cancer cells again to promote cancer progression, or the elevated ECM1 in exosome is just a reflection of ECM1 expression.

Minor concerns:

- 1) Fig. 1a,1e, the images for exosome are very poor quality and the bar should be shown more clearly.
- 2) Fig. 1b. protein names should be labeled.
- 3) Fig. 1f. not sure what it shows and please have clear labeling.
- 4) Fig. 1d, if the complement and coagulation cascade is the most elevated pathway in exosome, should it be the major focus of research?
- 5) Fig. 1h. Why have human and mouse exosomes such low overlapping proteins?
- 6) Fig. 2c. The picture has very low resolution and please replace 2b, 2c, 2d
- 7) Fig. 4a-b. Very poor quality.
- 8) Fig. 6f. magnification and bar should be shown.
- 9) Fig. S1d, a positive control should be included.
- 10) Fig. S5i. The MMP3 blot is very blurry.
- 11) Some minor grammar/spelling etc:
 - abstract, last sentence: exosomal-based should be “exosome-based”
 - introduction: first paragraph last sentence. “whether exosomes contributed to the enhanced BC metastasis and growth under obesity conditions has not been explored”. Should be changed to “whether exosomes contribute.....”
 - introduction: 2nd paragraph, please define “BMI quartile 4 v 2”; last sentence: please change “consider” to “considered”.
 - please carefully go through the grammars.

Reviewer #2 (Remarks to the Author): with expertise in exosomes, cancer

In this manuscript entitled “Circulating exosomal ECM1 protein level is elevated in association with integrin- β 2 and mediates the enhanced breast cancer growth and metastasis under obesity conditions”, the authors propose a novel mechanism of breast cancer metastasis by ECM1 proteins in the extracellular vesicles (EVs). The loading of ECM1 protein into the EVs was associated with integrin- β 2 levels in the donor cells. This manuscript provides an important contribution to the rapidly growing field of EVs; however, the quality of this manuscript is not enough for publication in its current form. Specific concerns are shown below.

Major

#1. It is challenging to distinguish between small vesicles using the current technologies because small vesicles derived from biological sources, such as exosomes (diameter: 50–200 nm), microvesicles (diameter: 100–1000 nm), apoptotic bodies (diameters 100-5000 nm), and other types of vesicles, are complex. Thus, the term "EVs" will be used by the International Society for Extracellular Vesicles (ISEV) to refer to all varieties of vesicles that exist in the extracellular space (Théry C, et al., JEV, 2018).

#2. Almost all the bands of WB are low quality. Especially, the difference of ECM1 reported here is marginal.

#3. All imaging data are of poor quality. The authors should change more clear ones. In addition to this, use larger fonts for the reviewers and readers.

#4. In the material and methods section, why did the authors use different EV collection methods for serum and plasma? Different methods yield different EVs that can be recovered.

#5. Product number is lacking in the methods, such as antibodies.

#6. The method of statistics is wrong (Figure 6b, 6d, 6e and 6f). It is not appropriate to evaluate with the student's t-test. The authors should re-calculate the p-value. Also, the authors should add the information (a one-sided or two-sided test)

#7. In the Figure S1b and S1e, why are the bands different sizes?

#8. In the Figure 1j, how did the authors exclude EVs?

#9. In the Figure 2f, why is ECM1 expression different between samples?

#10. In the Figure 2b-d, the authors showed the data of single cell analysis. These data are just pasted from Human Atlas. The image quality is poor, and nothing can be seen, so it should be reanalyzed. Macrophages should be pointed out in red.

#11. In the Figure 2a, the quantification does not reflect the actual band intensities. The reviewer doesn't know why the Mean \pm SEM match up in the C-SExos group.

#12. The same concern applies to Fig 4f. The authors counted the cell number of invaded cells. The authors should increase the field of view that counts. This experiment is not very convincing as the representativeness of this finding is unclear.

#13. The authors described that "Taken together, our data not only demonstrated that the enhanced exosomal ECM1 protein level was associated with integrin- β 2 expression, but also suggest monocytes/macrophages and adipocytes are the potential donor cells that release exosomes with elevated ECM1 protein levels under obesity conditions." Which cell-derived EVs contribute more to the cancer cells they give?

#14. In the Figure 4g, the authors should add the loading control like CD63.

#15. In the Figures 5a and 6a, the authors showed the scheme of animal studies. Why administer cancer cells to recover EVs (Figure 5a)? Why not do that in 6a? The authors should match the experimental methods.

#16. In the Figure S6a, the intensity of the background signal varies from photo to photo. Background intensities must be aligned.

#17. In the Figure 5b, the authors described that "Figure 5b showed that the tail-vein injected labeled exosomes could reach the tumors in the mammary pads in the mice." Why can the authors say that?

#18. In Figure 5g, the lung metastasis sites are counted. An image of HE staining of the lung metastasis sites should be added.

Minor

#1. Scale bar is too difficult to see in Figure 1a, 1c, 4a, 4b, 5d, 6c, 6f, S5g, S5h, S6a, S7d, S8b, S8e, S9d and S9g. Please check and add the size in the figure legend, respectively.

#2. In the Figure 1j, the characters of Exo-depleted text is missing.

#3. In the Figure S1a-d, what do the 30S, 150s, 60S and 3S mean? Please add the figure legend.

#4. In the Figure 2b, Y-axis is missing.

#5. Methods section (Single cell analysis of ECM1 and Itgb2), "Itgb2" should be "ITGB2"

#6. The authors may want to consider rearranging the figures such that they match the order of their descriptions in the text. For instance, Fig. S4a is called before Fig. S3c-f.

#7. In the Figure 3b, is the molecular weight of CD63 correct at 43 kDa?

#8. In the Figure 3b, p-value is missing.

Reviewer #3 (Remarks to the Author): with expertise in proteomics, cancer

In the underlying article "Circulating exosomal ECM1 protein level is elevated in association with integrin- β 2 and mediates the enhanced breast cancer growth and metastasis under obesity conditions", the authors describe that extracellular matrix protein 1 (ECM1) is transported by serum "exosomes" and is elevated under obesity conditions. They hypothesise that increased "exosomal" ECM1 levels promote breast cancer metastasis and growth, providing a new mechanism and target for treatment. This new potential mechanism should help explain the impact of obesity on breast cancer development.

The authors use different proteomic approaches (iTRAQ, MRM) to identify and quantify „exosomal“ proteins from human and mouse serum. They point to monocytes/macrophages and adipocytes as

potential donors of these „exosomes“. Overexpression of ECM1 in human breast cancer cells increased cell invasion/migration and promoted the expression of MMP3 and S100A/B. They then incubated breast cancer cells with purified, labelled „exosomes“, resulting in increased cellular levels of ECM1. Injection of purified „exosomes“ from affected mice leads to an increase in tumour size/weight and lung metastases. Knockout of Rab27a, which reduces exosome secretion, protected mice on HFD from increased tumour growth. Loading „exosomes“ with the ECM1 plasmid induced tumour growth and metastasis.

Overall, the experimental strategy is convincing and addresses an important medical problem. However, there are some issues and inconsistencies, especially in sample preparation and protein profiling.

„Exosomes“ from serum were enriched by centrifugation, while „exosomes“ from supernatants and adipose tissue were enriched using a kit. There is no other specific purification. Although these methods are widely used, there is a consensus that the yield and purity of the exosomes obtained is low. It would be more appropriate to speak of small extracellular vesicles rather than sticking to the term exosomes. Enrichment was monitored by TEM and particle size was determined by NTA for human serum „exosomes“ and by DLS for mouse serum „exosomes“. Why did the authors use different methods for enrichment and particle size determination in the same study? It would also be beneficial to provide larger sections of the TEM images and generally check the figures for readability (e.g. axis labelling of the DLS graph).

The authors state that they used iTRAQ-based quantitative proteomics. In the Methods section, this was mentioned again in the section heading. In contrast, the authors describe a label-free quantitative approach in the methods text. Furthermore, there is no information on the number of samples used for proteomic profiling. In the case of MRM, there is no information on the mass spectrometric parameters and the selected peptides used for quantification, nor on the software used.

The supplementary tables partly contain information in Chinese, which I unfortunately cannot read.

Point-to-Point Response to Editor's and Reviewers' comments

REVIEWER COMMENTS

Reviewer #1:

Circulating exosomal ECM1 protein level is elevated in association with integrin- β 2 and mediates the enhanced breast cancer growth and metastasis under obesity conditions by Xu et al.

This work by Xu et al described circulating exosomal ECM1 is elevated in obesity, contributing to breast cancer growth and metastasis. There is a correlation between exosomal ECM1 and integrin- β 2 co-loading into the exosomes. The origin of exosomes can be from monocytes/macrophages or adipocytes induced under obese conditions. Forced loading of ECM1 in exosomes can promote tumor growth and metastasis of breast cancer models. This project is interesting and has some potential impact to promote mechanistic understanding of how obesity promotes breast cancer progression and metastasis. Though this work has intrinsic innovation at the conceptual level, this manuscript, however, has some major issues related to data quality, in depth mechanisms and the significance in breast cancer relevance.

Major concerns:

1) *All data quality is very low and the legends in the pictures are illegible, examples including all bar graphs, images, and many other labels.*

Response:

Thank you for the comments. We have replaced the pictures, graphs and images with better quality, they are Figure 1a to 1i, Figure 2a to 2i, Figure 3a to 3d, Figure 4a to 4g, Figure 5a to 5g, and Figure 6a to 6f, supplementary Figures S7a to S7j, S8a to S8d, S9d to S9h, S10a, S10b, S10e, S11a to S11d and S11g.

2) *The major animal experiments are all with 3 mice per group; it is way under powered. For any tumor studies, even with 1.5 variations in volume between groups and with 0.8 power, a minimal of 7-8 mice per group should be required. For example, images shown in Figure 6C seem to have much bigger than 1.5 variations and the statistics in figure 6d did not recapitulate such variations.*

Response:

Thank you for the comments. We repeated the study with more mice in each group. The results are reported in Figures 5 and 6, in which injection of H-sEVs significantly increased tumor size, tumor weight and metastasis (Figure 5); while D-sEVs and ECM-sEVs treatments significantly increased tumor size, tumor weight and metastasis (Figure 6). The same findings were obtained in these repeated studies. We have highlighted the figure numbers in red in the revised manuscript.

3) *The pathological connection between ECM1 and breast cancer metastasis is established by using whole tumor RNAseq dataset, with most ECM1 transcripts expected to be expressed by cancer cells. Cancer cells overexpressing ECM1 also exhibit more tumor growth and metastasis. How much ECM1 can be achieved under physiological conditions to induce breast cancer progression and metastasis? ECM1 is expected a secreted protein working with cancer cells within the tumor microenvironment, it is logic to presume that most microenvironmental ECM1 comes from cancer cells or stromal cells, other than from exosomes. Figure S7b showed HFD increases ECM1 expression in tumors, but how can the authors prove those are secreted from tumors into the microenvironment or those come from serum exosomes?*

Response:

Thank you for the variable comments. We agree with Reviewer's comment that cancer cells express ECM1. Based on our data, we suggest that under obesity or HFD feeding conditions, breast cancer has higher ECM1 protein level, which is mainly due to the elevated ECM1 protein in the circulating sEVs that deliver the protein to the cancer cells. Our interesting data with Rab27a knockout mouse model (B6/J-Rab27a-Cas9-KO) showed that obesity failed to increase tumor ECM1 protein levels (supplementary Figure S9h), suggesting an important role of the sEVs in enhancing tumor ECM1 protein under obesity.

Based on the Reviewer's comment that ECM1 is a secretory protein, we have done additional experiments to suggest after D-sEVs treatments that deliver more ECM1 protein to the cancer cells, these cells secrete more ECM1 protein. As shown in Figure 4c, treatments with D-sEVs significantly increased the ECM1 protein levels in the E0771 cells compared to C-sEVs treatments. Besides, the D-sEVs-treated cells also secreted significantly more ECM1 protein compared to C-sEVs-treated cells (Figure 4f).

In the animal study, we agree with Reviewer's comment that the Western blot analysis presented in Figures 5c and 6b does not allow for differentiation between ECM1 protein present in tumor cells, ECM1 secreted into the tumor microenvironment, or ECM1 found in the sEVs within tumor blood vessels. Indeed, the enhanced delivery of ECM1 through sEVs to the tumor, uptake of ECM1 by tumor cells from sEVs, and the subsequent secretion of ECM1 into the tumor microenvironment are interrelated and consequential events. Therefore, in the revised manuscript, we have corrected our interpretation of the data to "injection of H-sEVs increases tumor ECM1 levels"; and we also clarified that "The observed increase in ECM1 levels within the tumor can be attributed to multiple factors, including the enhanced delivery of ECM1 by sEVs to the tumor, increased release of ECM1 from tumor cells, and elevated levels of ECM1 present in sEVs within the tumor microenvironment".

Under non-obese conditions, ECM1 also promotes breast cancer progression and metastasis. In breast cancer, ECM1 has been implicated in the control of cancer stem cell-like properties and the process of epithelial-to-mesenchymal transition by stabilizing β -catenin (Lee, 2015a). Notably, ECM1 also contributes to the metastasis of aggressive breast cancer by regulating actin cytoskeletal architecture

(Gomez-Contreras, 2017). Besides, ECM1 induces angiogenesis in BC (Colomer R, 2000). Furthermore, ECM1 plays a role in promoting resistance to trastuzumab and inducing the pyruvate kinase M2-mediated Warburg effect by activating epidermal growth factor signaling pathways (Lee, 2014; Lee, 2015b). This information has been included in the Discussion in the revised manuscript, highlighted in red.

Please kindly let us know whether we have addressed your concern in this question or we can revise it and provide further information. Thank you very much.

References

- Colomer, R., Montero, S., Lluch, A., Ojeda, B., Barnadas, A., Casado, A., Massuti, B., Cortés-Funes, H., Lloveras, B. (2000) Circulating HER2 extracellular domain and resistance to chemotherapy in advanced breast cancer. *Clin Cancer Res*, 6, 2356–62.
- Gomez-Contreras, P., Ramiro-Díaz, J.M., Sierra, A., Stipp, C., Domann, F.E., Weigel, R.J., Lal, G. (2017) Extracellular matrix 1 (ECM1) regulates the actin cytoskeletal architecture of aggressive breast cancer cells in part via S100A4 and Rho-family GTPases. *Clin Exp Metastasis*, 34, 37–49.
- Lee, K. M., Nam, K., Oh, S., Lim, L., Kim, Y., Lee, J.W., Yu, J., Ahn, S., Kim, S., Noh, D., Lee, T., Shin, I. (2014) Extracellular matrix protein 1 regulates cell proliferation and trastuzumab resistance through activation of epidermal growth factor signaling. *Breast Cancer Res*, 16, 479.
- Lee, K. M., Nam, K., Oh, S., Lim, J., Kim, R.K., Shim, D., Choi, J., Lee, S., Yu, J., Lee, J., Ahn, S.H., Shin, I. (2015a) ECM1 regulates tumor metastasis and CSC-like property through stabilization of beta-catenin. *Oncogene*, 34, 6055–65.
- Lee, K. M., Nam, K., Oh, S., Lim, J., Lee, T., Shin, I. (2015b) ECM1 promotes the Warburg effect through EGF-mediated activation of PKM2. *Cell Signal*, 27, 228–35.

4) *The study lacks of mechanistic explanation for exosomal ECM1 to work in cancer cells. ECM1 is a secreted protein and mostly known for its function. Cancer cells pick exosomes and take the exosomal proteins within cells. Does ECM1 from exosome need to be secreted by cancer cells again to promote cancer progression, or the elevated ECM1 in exosome is just a reflection of ECM1 expression.*

Response: Thank you for the comment. Yes, based on the Reviewer’s comment that ECM1 is a secretory protein that acts on the cells to exert its function, we have done additional experiments in the revised manuscript in which after D-sEVs treatments, E0771 cells secreted significantly more ECM1 protein into the extracellular matrix compared to cells treated with C-sEVs (Figure 4f). Moreover, the extracellular matrix collected from D-sEVs-treated cells significantly promoted cancer cell invasion and migration compared to the matrix derived from C-sEVs-treated cells (Figure 4g). To investigate whether the elevated invasion and migration were indeed a result of the increased ECM1 protein in the extracellular matrix, we

used immunoprecipitation to selectively deplete ECM1 protein from the matrix (Figure 4f) before using it for treatments. As shown in Figure 4g, after ECM1 was depleted from the extracellular matrix derived from the D-sEV-treated cells, the treatments failed to increase cell invasion and migration. These findings collectively suggest that D-sEVs increase the delivery of ECM1 protein to cancer cells, and these cells increase ECM1 secretion. The secreted ECM1 acts on the cancer cells to promote invasion and migration. The results are highlighted in red in the revised manuscript.

Minor concerns:

1) *Fig. 1a,1e, the images for exosome are very poor quality and the bar should be shown more clearly.*

Response:

Yes, we have replaced the images with better quality and clear bars. The figure numbers are highlighted in red.

2) *Fig. 1b. protein names should be labeled*

Response:

We have labeled the protein names in the revised Figure 1b.

3) *Fig. 1f. not sure what it shows and please have clear labeling.*

Response:

Figure 1f in the original manuscript was the DLS study on the purified mouse sEVs, which can be also used to measure the diameter of particles. To keep the monitoring methods consistent, we have reexamined the purified mouse sEVs with NTA shown in Figure 1e in the revised manuscript. The figure number is highlighted in red.

4) *Fig. 1d, if the complement and coagulation cascade is the most elevated pathway in exosome, should it be the major focus of research?*

Response: Thank you for the comment. Yes, complement and coagulation cascade is the most elevated pathway in the analysis. Here, we focus to examine whether sEVs play a role in cancer metastasis. Among the top four highlighted pathways, it is known that ECM-receptor interaction contributes to the metastasis of BC (Gómez-Contreras, 2017). Therefore, in this study, we focused on the ECM signaling pathway.

Reference:

Gómez-Contreras, P., Ramiro-Díaz, J.M., Sierra, A., Stipp, C., Domann, F.E., Weigel, R.J., Lal, G. (2017) Extracellular matrix 1 (ECM1) regulates the actin cytoskeletal architecture of aggressive breast cancer cells in part via S100A4 and Rho-family GTPases. *Clin Exp Metastasis* 34(1), 37-49.

5) *Fig. 1h. Why have human and mouse exosomes such low overlapping proteins?*

Response:

We only compared the sEVs proteins that were differentially expressed with statistically significant difference between (1) human obesity and healthy control samples, and (2) DIO and control diet mouse samples. Therefore, our data show that there is a total of 15 overlapping DEPs (Figure 1g).

6) *Fig. 2c. The picture has very low resolution and please replace 2b, 2c, 2d*

Response:

Yes, we have replaced Figures 2b, 2c and 2d with figures of better resolution. The figure numbers are highlighted in red in the revised manuscript.

7) *Fig. 4a-b. Very poor quality.*

Response:

Yes, we have repeated the study and replaced the figures with better quality in the revised manuscript. The figure numbers are highlighted in red.

8) *Fig. 6f. magnification and bar should be shown.*

Response:

Yes, we have included magnification and bar chart in Figure 6f and also Figure 5g.

9) *Fig. S1d, a positive control should be included.*

Response:

Yes, we have included a cell sample as positive control, which showed the expression of calnexin. The figure number is highlighted in red in the revised manuscript.

10) *Fig. S5i. The MMP3 blot is very blurry.*

Response:

We have repeated the study and replaced the blot, which is supplementary Figure S7i in the revised manuscript. The figure number is highlighted in red.

11) *Some minor grammar/spelling etc:*

Response:

Thank you for your comment. We have corrected the grammatical and spelling mistakes.

- *abstract, last sentence: exosomal-based should be “exosome-based”*

Response:

We have corrected the mistake, and it is now written as “sEV-based”.

- *introduction: first paragraph last sentence. “whether exosomes contributed to the enhanced BC metastasis and growth under obesity conditions has not been explored”. Should be changed to “whether exosomes contribute.....”*

Response:

Thank you very much. We have corrected the mistakes.

- *introduction: 2nd paragraph, please define “BMI quartile 4 v 2”; last sentence: please change “consider” to “considered”.*

Response:

In the cited study (Goodwin, 2012), BMI-related variables were entered as continuous variables into the Cox models and hazard ratios (HRs) calculated at the midpoints of the quartiles (i.e., 12.5th, 37.5th, 62.5th, and 87.5th percentiles). We have revised the writing to “Obesity-related variables such as body mass index (BMI) and weight exert significant adverse univariable associations with BC distant recurrence over time. BMI is modeled quadratically, for the quartiles 4 vs. quartile 2: hazard ratio, 1.40; 95% CI, 1.07 to 1.82 for distant disease-free survival; $P < .014$; and hazard ratio, 1.50; 95% CI, 1.16 to 1.93; $P < .001$ for overall survival”. The revised writing is highlighted in red.

Reference:

Goodwin, P.J., Ennis, M., Pritchard, K.I., Trudeau, M.E., Koo, J., Taylor, S.K., Hood, N. (2012) Insulin- and obesity-related variables in early-stage breast cancer: Correlations and time course of prognostic associations. *J Clin Oncol*, 30, 164-71.

- *please carefully go through the grammars.*

Response:

Thank you very much. We have corrected the grammatical mistakes.

Reviewer #2:

In this manuscript entitled “Circulating exosomal ECM1 protein level is elevated in association with integrin- β 2 and mediates the enhanced breast cancer growth and metastasis under obesity conditions”, the authors propose a novel mechanism of breast cancer metastasis by ECM1 proteins in the extracellular vesicles (EVs). The loading of ECM1 protein into the EVs was associated with integrin- β 2 levels in the donor cells. This manuscript provides an important contribution to the rapidly growing field of EVs; however, the quality of this manuscript is not enough for publication in its current form. Specific concerns are shown below.

Major

#1. It is challenging to distinguish between small vesicles using the current technologies because small vesicles derived from biological sources, such as exosomes (diameter: 50–200 nm), microvesicles (diameter: 100–1000 nm), apoptotic bodies (diameters 100–5000 nm), and other types of vesicles, are complex. Thus, the term "EVs" will be used by the International Society for Extracellular Vesicles (ISEV) to refer to all varieties of vesicles that exist in the extracellular space (Théry C, et al., JEV, 2018).

Response: Thank you for the comment. We agree with the Reviewer’s comment. We have used the term sEV in the revision.

#2. Almost all the bands of WB are low quality. Especially, the difference of ECM1 reported here is marginal.

Response: Thank you for the comment. We have repeated all the WB analysis in the revision, and replaced the blots in the figures, they are Figures 1h, 1i, 2a, 2e to 2i, 3a to 3d, 4c, 5c, 6b, supplementary Figures S7f, S7i, S8b, S9h, S10a, S11a to S11c. The figure numbers are highlighted in red.

#3. All imaging data are of poor quality. The authors should change more clear ones. In addition to this, use larger fonts for the reviewers and readers.

Response:

Thank you for the comment. We have repeated all the imaging study in the revision, and replaced the images in Figures 4a, 4b, 4e, 4g, 5g, 6f, supplementary Figures S7g, S7h, S8a, S8d, S9g, S10e and S11g. The figure numbers are highlighted in red.

#4. In the material and methods section, why did the authors use different EV collection methods for serum and plasma? Different methods yield different EVs that can be recovered.

Response:

Thank you for the comment. We have revised it in the revision, Differential centrifugations were used to isolate the sEVs from human and mouse plasma. Human plasma (4 mL) was diluted with 30 mL PBS, and mouse plasma (0.5 mL) were diluted with 1 mL PBS. Then, the diluted plasma was centrifuged to get supernatant at 2,000 x g for 10 min, and 10,000 x g for 30 min at 4 °C to eliminate dead cells and cell debris. Then they were centrifugated at 160,000 x g for 1 h (Beckman Coulter) twice to get the pellets that were resuspended in PBS followed by filtration with 0.22 um membrane filters. Finally, the pellets resuspensions were centrifuged at 160,000 x g for 1 hr to get the sEVs. The revision is highlighted in red.

#5. Product number is lacking in the methods, such as antibodies.

Response:

We have included the product numbers in the revised manuscript.

#6. The method of statistics is wrong (Figure 6b, 6d, 6e and 6f). It is not appropriate to evaluate with the student's t-test. The authors should re-calculate the p-value. Also, the authors should add the information (a one-sided or two-sided test)

Response: Thank you for the comment. We have revised it. The Student's t-test (two-sided, unpaired) was used for statistical evaluation for two group comparisons. Ordinary one-way ANOVA with Tukey's multiple comparison test for three group comparisons. All statistical analyses were performed with Prism software (GraphPad Prism for Windows, version 9.0). P value < 0.05 was considered statistically significant. The revision is highlighted in red in the revised manuscript.

#7. In the Figure S1b and S1e, why are the bands different sizes?

Response:

We used the anti-CD63 antibody from Abcam (#ab216130) for the Western blot analysis. With reference to the data sheet provided by the company, the predicted molecular size for CD63 is ~28kDa as shown in Figure A.

Figure A (it is the reference taken from the data sheet for #ab216130, Abcam).

(<https://www.abcam.com/products/primary-antibodies/cd63-antibody-late-endosome-marker-ab216130.html>).

In fact, CD63 has a number of potential glycosylation sites that may affect the migration of the protein. Figure B shows the CD63 glycosylation in breast cancer cells is associated with RPN2 (Tominaga, 2014). The molecular weight of glycosylated CD63 is reduced after N-glycanase treatment in the breast cancer cell lines. Inhibition of RPN2 expression reduces CD63 glycosylation as suggested by the reduced molecular weight of CD63 in RPN2 siRNA-treated cells compared to control siRNA-treated cells (Tominaga, 2014).

Figure B: Whole cell lysates were collected from MDA-MB-231-luc-D3H2LN (MM231-LN) (upper panel) and MCF7-ADR cells (lower panel) that were transiently transfected with N.C. or RPN2 siRNA and treated with PBS for 6 hours at 37°C. N.C. and RPN2 siRNA-transfected cells were treated with N-glycosidase for 6 hours at 37°C. CD63 glycosylation was detected by immunoblotting. β-actin was used as a loading control. The CD63 molecular weights of 25 (non-glycosylated), 35 (lower-glycosylated) and 50 kDa (higher-glycosylated) are indicated with arrows to the right. (Tominaga, 2014).

We also take reference from the company NOVUS that shows the size of CD63 varies from ~35 to 55 kDa (Figure C), or from ~25 to 50 kDa (Figure D). A study also shows the size of CD63 is between ~37 to 50 kDa (Figure E) (Olivero, 2021).

Figure C: It is a reference taken from NOVUS product CD63 Antibody (MX-49.129.5) [NBP2-32830]. Exosomes or whole cell lysates (WCL) sample from MDA-MB-231 cells was loaded with 10 μg/lane. 10% SDS-PAGE. CD63 antibody (NBP2-32830) was used for primary antibody: 1:500, 4C, overnight. (https://www.novusbio.com/products/cd63-antibody-mx-491295_nbp2-32830)

Figure D: A reference taken from the company for the antibody against CD63 (Novus Biologicals NBP27881220UG) (<https://www.fishersci.com/shop/products/cd63-mouse-anti-human-mouse-clone-lamp3-2788-novus-biologicals-3/NB002964>)

Figure E: Western blot analysis of proteins in the cortical synaptosomal and exosomal lysate. Identical amounts of the cortical synaptosomal lysate (lane Syn) and of the exosomal lysate (lane Exo) were loaded on 10% SDS-PAGE and analyzed for the contents of the exosomal markers including CD63, TSG101, CD9 and flotillin-1 and of the synaptosomal markers syntaxin-1a and PSD95 (Olivero, 2021).

We expect the difference sizes of CD63 in supplementary Figure S1b and S1e is due to the different glycosylation of the CD63 proteins in human and mouse.

References

- Olivero, G., Cisani, F., Marimpietri, D., Di Paolo, D., Gagliani, M.C., Podesta M., Cortese, K., Pittaluga, A. (2021) The depolarization-evoked, Ca²⁺-dependent release of exosomes from mouse cortical nerve endings: New insights into synaptic transmission. *Front Pharmacol*, 12, 670158.
- Tominaga, N., Hagiwara, K., Kosaka, N., Honma, K., Nakagama, H., Ochiya, T. (2014) RPN2-mediated glycosylation of tetraspanin CD63 regulates breast cancer cell malignancy. *Mol Cancer*, 13, 134.

#8. In the Figure 1j, how did the authors exclude EVs?

Response:

Thank you for the comment. Mouse plasma (1.9 mL) was centrifuged to get supernatant at 2,000 x g for 10 min, and 10,000 x g for 30 min at 4°C to remove dead cells and cell debris. Then they were centrifugated at 160,000 x g for 15 hrs (Beckman Coulter) and only supernatant was collected. CD63 expression was used as a marker to check the removal of sEV in the sample (supplementary Figure 1e). We have included the methodology accordingly, they are highlighted in red.

#9. In the Figure 2f, why is ECM1 expression different between samples?

Response:

Thank you for your comment. We have repeated the western blot analysis and Figure 2f has been updated. The results show that ECM1 expression is reduced in the sEVs derived from siRNA-integrin β 2-transfected RAW264.7 cells (si- β 2) when compared to that in the sEVs derived from control cells (NC). The figure number is highlighted in red in the revised manuscript.

#10. In the Figure 2b-d, the authors showed the data of single cell analysis. These data are just pasted from Human Atlas. The image quality is poor, and nothing can be seen, so it should be reanalyzed. Macrophages should be pointed out in red.

Response:

Thank you for your comment. We have reanalyzed the data and macrophages are pointed out in red in Figures 2c and 2d. Figures 2b, 2c and 2d are replaced with better quality images. The figure numbers are highlighted in red in the revised manuscript.

#11. In the Figure 2a, the quantification does not reflect the actual band intensities. The reviewer doesn't know why the Mean \pm SEM match up in the C-SExos group.

Response:

We have repeated the Western blot and the analysis, Figure 2a is updated in the revision. The result shows that integrin β 2 is significantly elevated in the sEVs of the DIO mice (D-sEVs). The figure number is highlighted in red in the revised manuscript.

#12. The same concern applies to Fig 4f. The authors counted the cell number of invaded cells. The authors should increase the field of view that counts. This experiment is not very convincing as the representativeness of this finding is unclear.

Response:

Thank you for the comment. We have repeated the study, and the figures are updated. Figure 4f in the original manuscript is presented as Figure 4e in the revision, in which the data clearly shows that D-sEVs treatments significantly increase E0771 cell invasion and migration. The figure number is highlighted in red in the revised manuscript.

#13. The authors described that "Taken together, our data not only demonstrated that the enhanced exosomal ECM1 protein level was associated with integrin- β 2 expression, but also suggest monocytes/macrophages and adipocytes are the potential donor cells that release exosomes with elevated

ECM1 protein levels under obesity conditions.” Which cell-derived EVs contribute more to the cancer cells they give?

Response:

Thank you for the question. We found that sEVs derived from macrophages and adipocytes have elevated ECM1 protein level under obesity conditions. The macrophages and adipocytes also have elevated integrin- $\beta 2$ expressions that may increase ECM1 protein loading to the sEVs. Under obesity conditions, adipose tissues expand, and this expansion is accompanied by the accumulation of macrophages, which play a role in chronic inflammation (Weisberg, 2003). The contribution of sEVs derived from adipocytes and macrophages to BC can be influenced by various factors, including abundance of adipocytes and the inflammatory state associated with the number of macrophages. Release of sEVs by these cell types may vary in response to different impacts of obesity on the pathological conditions. Studies have shown that adipose tissues release a higher quantity of sEVs under obesity conditions (Son, 2023). Therefore, it is plausible to speculate that both macrophages and adipocytes play significant roles in delivering sEVs containing elevated levels of ECM1 to the breast tumor. However, in this study, we cannot compare their relative contribution to the BC. The manuscript is revised accordingly, and the revision is highlighted in red.

Reference

Weisberg, S.P., McCann, D., Desai, M., Rosenbaum, M., Leibel, R.L., Ferrante, A.W., Jr. (2003) Obesity is associated with macrophage accumulation in adipose tissue. *J Clin Investig*, 112, 1796–1808.

Son, T., Jeong, I., Park, J., Jun, W., Kim, A., Kim, O.K. (2023) Adipose tissue-derived exosomes contribute to obesity-associated liver diseases in long-term high-fat diet-fed mice, but not in the short-term. *Front Nutr*, 10, 1162992.

#14. In the Figure 4g, the authors should add the loading control like CD63.

Response:

Thank you for the comment. Since ECM1 is a secreted protein, we have revised the method by using ELISA to examine the ECM1 secreted by the BC cells after c-sEVs or D-sEVs treatments. The original Figure 4g is presented as Figure 4f in the revision. The revised figure number is highlighted in red.

#15. In the Figures 5a and 6a, the authors showed the scheme of animal studies. Why administer cancer cells to recover EVs (Figure 5a)? Why not do that in 6a? The authors should match the experimental methods.

Response:

Thank you for the Reviewer's comment. We have revised the protocol for these studies in the revision. In the study for Figure 5, we purified sEVs from the plasma of HFD and CD-fed BALB/c mice. In the study for Figure 6, we purified sEVs from the plasma of DIO and CD-fed C57BL/6 mice. These purified sEVs were then injected into the respective mouse models for the examination of their effects on BC growth and metastasis. The figure numbers for the schematic designs (Figure 5b and 6a) are highlighted in red in the revision.

#16. In the Figure S6a, the intensity of the background signal varies from photo to photo. Background intensities must be aligned.

Response:

We have repeated the study. Supplementary Figure S6a in the original manuscript is presented as supplementary Figure S8a in the revision. The photos are updated in which we have aligned the background intensities and included the bar. The figure number is highlighted in red in the revision.

#17. In the Figure 5b, the authors described that "Figure 5b showed that the tail-vein injected labeled exosomes could reach the tumors in the mammary pads in the mice." Why can the authors say that?

Response:

Thank you for the comment. In this experiment, we established the mouse model by injecting 4T1 cells into the mammary fat pad (Figure 5a, red arrow in the upper panel in the revision). Then we injected the DIR-labeled sEVs via the tail vein of the mouse. As shown in Figure 5a (lower panel), the signal could be detected in the tumor site, suggesting the tail-vein injected labeled sEVs are accumulated in the tumors of the mice. Figure 5b in the original manuscript is presented as Figure 5a in the revision. We have revised the writing accordingly, which is highlighted in red.

#18. In Figure 5g, the lung metastasis sites are counted. An image of HE staining of the lung metastasis sites should be added.

Response:

Thank you for the comment. We have done the HE staining of the lung metastasis in the revision shown in Figure 5g. The result shows that H-sEV treatments significantly increased the cancer lung metastasis. The figure number is highlighted in red in the revision.

Minor

#1. Scale bar is too difficult to see in Figure 1a, 1c, 4a, 4b, 5d, 6c, 6f, S5g, S5h, S6a, S7d, S8b, S8e, S9d and S9g. Please check and add the size in the figure legend, respectively.

Response:

We have improved them in the revised manuscript. The figure numbers are highlighted in red in the revision.

#2. In the Figure 1j, the characters of Exo-depleted text is missing.

Response:

Thank you for the comment. Figure 1j is now presented as Figure 1i in the revision and we have corrected the mistake.

#3. In the Figure S1a-d, what do the 30S, 150s, 60S and 3S mean? Please add the figure legend.

Response:

“30S, 150s, 60S and 3S” are the exposure time in the Western blot analysis. We are sorry for the mistakes, and we have deleted them as they are irrelevant to the results.

#4. In the Figure 2b, Y-axis is missing.

Response:

We have corrected the mistake in the revision. The Y-axis is $-\log_{10}$ (p-value). The figure number is highlighted in red in the revision.

#5. Methods section (Single cell analysis of ECM1 and Itgb2), “Itgb2” should be “ITGB2”

Response:

Thank you for the comment. We have corrected the mistakes.

#6. The authors may want to consider rearranging the figures such that they match the order of their descriptions in the text. For instance, Fig. S4a is called before Fig. S3c-f.

Response:

Thank you for the comment. We have rearranged the order of the figures that match the descriptions in the revised manuscript.

#7. In the Figure 3b, is the molecular weight of CD63 correct at 43 kDa?

Response:

We have corrected the “typo” mistake. The size of CD63 in Figure 3b should be 26 kDa. The figure number is highlighted in red in the revision.

#8. In the Figure 3b, p-value is missing.

Response:

Thank you for the comment. We have included the p-values in the revision. The figure numbers are highlighted in red in the revised manuscript.

Reviewer #3

In the underlying article "Circulating exosomal ECM1 protein level is elevated in association with integrin- β 2 and mediates the enhanced breast cancer growth and metastasis under obesity conditions", the authors describe that extracellular matrix protein 1 (ECM1) is transported by serum "exosomes" and is elevated under obesity conditions. They hypothesized that increased "exosomal" ECM1 levels promote breast cancer metastasis and growth, providing a new mechanism and target for treatment. This new potential mechanism should help explain the impact of obesity on breast cancer development.

The authors use different proteomic approaches (iTRAQ, MRM) to identify and quantify "exosomal" proteins from human and mouse serum. They point to monocytes/macrophages and adipocytes as potential donors of these "exosomes". Overexpression of ECM1 in human breast cancer cells increased cell invasion/migration and promoted the expression of MMP3 and S100A/B. They then incubated breast cancer cells with purified, labelled "exosomes", resulting in increased cellular levels of ECM1. Injection of purified "exosomes" from affected mice leads to an increase in tumour size/weight and lung metastases. Knockout of Rab27a, which reduces exosome secretion, protected mice on HFD from increased tumour growth. Loading "exosomes" with the ECM1 plasmid induced tumour growth and metastasis.

Overall, the experimental strategy is convincing and addresses an important medical problem. However, there are some issues and inconsistencies, especially in sample preparation and protein profiling.

"Exosomes" from serum were enriched by centrifugation, while "exosomes" from supernatants and adipose tissue were enriched using a kit. There is no other specific purification. Although these methods are widely used, there is a consensus that the yield and purity of the exosomes obtained is low. It would be more appropriate to speak of small extracellular vesicles rather than sticking to the term exosomes.

Response:

Thank you for the comment. Yes, we agree with Reviewer's comment. We have revised the manuscript and used the term "sEV" instead of exosomes.

Enrichment was monitored by TEM and particle size was determined by NTA for human serum "exosomes" and by DLS for mouse serum "exosomes". Why did the authors use different methods for enrichment and particle size determination in the same study?

Response:

Thank you for the comment. We have repeated the examination and used NTA for both human and mouse sEVs in the revision. The results show that the sizes of the purified sEVs are ~100nm. The figure numbers (Figure 1a, 1e and supplementary Figure 4b) are highlighted in red in the revised manuscript.

It would also be beneficial to provide larger sections of the TEM images and generally check the figures for readability (e.g. axis labelling of the DLS graph).

Response:

We have repeated the TEM study and replaced the photos with larger section and better quality in Figures 1a, 1e and supplementary figure S4a. The figure numbers are highlighted in red in the revised manuscript.

The authors state that they used iTRAQ-based quantitative proteomics. In the Methods section, this was mentioned again in the section heading. In contrast, the authors describe a label-free quantitative approach in the methods text.

Response:

Thank you for the comment. We have corrected the mistakes. iTRAQ-based quantitative proteomics was used to examine the protein profiles of mouse sEV samples. Protein samples were enzymatically hydrolyzed by trypsin. After enzymatic hydrolysis, the protein was cleaved into small peptides for subsequent mass spectrometry analysis. Peptides in different samples were labeled separately with ITRAQ reagent. ITRAQ reagent is a chemical reagent containing isotope markers that reacts with amino groups in peptides to form labeled peptides. The labeled peptides were mixed to form a composite sample. Peptides in complex samples were separated by high performance liquid chromatography (HPLC, ThermoScientific). The separated peptides were analyzed on Orbi MS (1-3ppm), Orbi MS/MS. With mass spectrometry analysis, the mass spectral peak of the peptide was obtained, and its intensity was measured. The mass spectrometer acquired raw mass spectrometry data in RAW format, and then used Proteome Discoverer 1.4 (Version 1.4.0.288, Thermo Fisher) to convert the RAW file to an MGF format mass spectrometry file and entered the MGF format mass spectrometry file and protein retrieval library into ProteinPilot™ Software 4.5 (version 1656, AB Sciex) for mass spectrometry search. The original data retrieved by the ProteinPilot software was analyzed, and the analysis process: 1) After the retrieval was completed, filter the Unused value for the original search results, and set Unused \geq 1.3 to make the protein trustworthy. The degree is above 95%; 2) To remove the anti-database in the search results, and to remove the records starting with "RRRRR" in the search results. ProteinPilot™ was performed FDR analysis. The protein data obtained by the above step analysis was a trusted protein. Differential screening for trusted proteins, screening process: 1) remove proteins without quantitative information; 2) Number of peptides \geq 2; 3) The mean value of the data ratio of each group was calculated, and BT: BC \geq 1.3 was selected as the upregulated protein and BT: BC \leq 0.77 was the downregulated protein; The data obtained by the above step analysis were differentially expressed proteins. The methodology is highlighted in red in the revision.

Furthermore, there is no information on the number of samples used for proteomic profiling.

Response:

We have included detailed description in the method section in the revision, highlighted in red. We recruited a total of 48 healthy people (11 males and 37 females) with $BMI \leq 23$ and without acute and chronic diseases in the normal weight healthy control group. A total of 96 people (77 males and 19 females) with $BMI \geq 25$ and without any infectious diseases, cancer, surgery, and genetic diseases were included in the obese group. A total of 0.5 mL of the EDTA anticoagulant plasma was taken from each participant in the clinical tests. Plasma of 8 participants were combined into one sample tube with a total of 4 mL in volume. A sample tube in each group was used for sEVs extraction for subsequent studies, the remaining sample tubes were used for proteomics study. For the mouse sEV proteomics study, a total of 4 female and 4 male C57BL/6 mice were included in control diet group (BC); a total of 4 female and 4 male C57BL/6 mice were included in high fat diet group (BT). Multiple ultracentrifugation was used to purify sEVs from mouse plasma. The methodology is highlighted in red in the revision.

In the case of MRM, there is no information on the mass spectrometric parameters and the selected peptides used for quantification, nor on the software used.

Response:

Thank you for the comment. We have included the details for MRM in the method section in the revision, highlighted in red. A total of 30 selected protein expressions were examined by MRM; the information of the proteins and peptide sequences are listed in supplementary Table S3. The MRM raw data of all samples were imported into Skyline software for quantitative statistics. In order to improve the quantitative statistical accuracy and signal-to-noise ratio (S/N), statistical correction is performed according to the following principles: a) Select at least one peptide to calculate the quantitative result of a target protein; b) Select at least three transitions to calculate the quantitative results of the target peptide; c) Manually remove any overlapping transitions with saturated signal-to-noise ratio <5 . The quantitative result of MRM statistics was based on the peak area (area) of the chromatographic peak of the ion. The quantitative value of the target protein was calculated by weighting the total area of several polypeptides of the protein. The total area of the target peptide was calculated by weighting the fragment ion area of the peptide.

The supplementary tables partly contain information in Chinese, which I unfortunately cannot read.

Response:

We are sorry for the mistake. We have translated the Chinese into English in the supplementary tables in the revision.

REVIEWERS' COMMENTS

Reviewer #1 (Remarks to the Author):

Thanks for the thorough revision. It is improved based on my previous comments. There are some minor issues:

1) Quality of figures are still low for many of them. Please make sure to relabel so that the legend is visible.

2) Quality for H&E staining is very poor and most of the arrows for lung metastasis don't seem like metastatic nodules. Please read by a pathologist to make sure the reading is correct.

Reviewer #2 (Remarks to the Author):

This revised version is much improved. The authors addressed almost all of my concerns.

Reviewer #4 (Remarks to the Author):

For the iTRAQ-based quantitative proteomics, the authors now state: "The separated peptides were analyzed on Orbi MS (1-3ppm), Orbi MS/MS." This is still very unclear as there are several types of mass spectrometers with Orbitrap analyzers. Moreover, the reader needs the basic settings used for the analysis, in the case of iTRAQ e.g. isolation width, fragmentation energy, etc. For the data analysis please report basic settings such as mass accuracy, missed cleavages and importantly for iTRAQ % of co-isolation allowed.

For the MRM, the authors now included some information on the data analysis using Skyline but all experimental settings such as MS instrument settings, LC gradient, etc. are still missing.

In the data availability it's stated:

"All data of the study is available from the corresponding authors upon request." This is not desirable for proteomics data and raw files with search output should be made available via a repository like Pride (ProteomeXchange).

Point-to-Point Response to Editor's and Reviewers' comments

REVIEWER COMMENTS

Reviewer #1

1) *Quality of figures are still low for many of them. Please make sure to relabel so that the legend is visible.*

Response: Thank you for the comment. We have improved the quality of figures 1b-1g; figures 4g, 4e; figure s5a-b, 5g; figures 6a, 6f; supplementary figures S9g, S10e, S11g. We also relabeled the legends of figures 1-6 and supplementary figures S1, S5, S7-12.

2) *Quality for H&E staining is very poor and most of the arrows for lung metastasis don't seem like metastatic nodules. Please read by a pathologist to make sure the reading is correct.*

Response: The metastatic nodules are marked by arrows by a pathologist. There are still some micro-metastatic nodules, which are not labeled in the H&E images but are included for statistical analysis.

Reviewer #2

This revised version is much improved. The authors addressed almost all of my concerns.

Response: Thank you for the nice comment.

Reviewer #4

1) *For the iTRAQ-based quantitative proteomics, the authors now state: "The separated peptides were analyzed on Orbi MS (1-3ppm), Orbi MS/MS." This is still very unclear as there are several types of mass spectrometers with Orbitrap analyzers. Moreover, the reader needs the basic settings used for the analysis, in the case of iTRAQ e.g. isolation width, fragmentation energy, etc. For the data analysis please report basic settings such as mass accuracy, missed cleavages and importantly for iTRAQ % of co-isolation allowed.*

Response: Thank you for the comment. We have included the information in the revised manuscript, labeled in red, and copied below:

Peptides in complex samples were separated by liquid chromatography (first dimension: LC-20AD, Shimadzu; second dimension: Dionex Ultimate 3000 RSLCnano, Thermo Scientific). The separated peptides were analyzed on Q Exactive Orbitrap Mass Spectrometers (MS, Thermo Scientific).

The basic settings of MS were: First-level mass spectrometry parameters: Resolution: 70,000; AGC target: 3e6; Maximum injection time: 100 ms; Scan range: 350 to 1800 m/z.

The secondary mass spectrometry parameters were: Resolution: 17,500; AGC target: 5e4; Maximum injection time: 120 ms; Top precursor ions: 20; Stepped Normalized Collision Energy (SNCE): 30.

The basic settings of data analysis were: Detected Protein Threshold [Unused ProtScore (Conf)] > 0.05(10%); Competitor Error Margin (ProtScore): 2; Paragon™ Algorithm: 4.5.0.0, 1654; Annotations Retrieved from UniProt; Sample Type: iTRAQ 8plex (Peptide Labeled); Cys. Alkylation: MMTS; Instrument: Orbi MS (1-3ppm), Orbi MS/MS.

2) For the MRM, the authors now included some information on the data analysis using Skyline but all experimental settings such as MS instrument settings, LC gradient, etc. are still missing.

Response: Sorry for the missing information. We have included the information in the revised manuscript, labeled in red.

The experimental settings of MRM were:

1) Liquid phase conditions:

Phase A was 2% ACN, 0.1% FA. Phase B was 98% ACN, 0.1% FA. The flow rate was 5 μ l/min. Gradient conditions were 0-5min, 95% A, 5% B; 5~105min, 70%, 30%B; 105~115min, 20% A, 80%B; 115~120min, 98% A, 2%B.

2) Mass spectrometry conditions:

The ion source: electrospray ion source (ESI); positive ion mode detection; Scanning mode: multiple reaction monitoring (MRM); injection voltage: 5500eV; temperature: 150°C; curtain gas (CUR, N₂): 30psi; collision gas pressure (CAD, N₂): High mode, the auxiliary gas GAS1 pressure=20psi; The auxiliary gas GAS2 pressure: 15psi; The scan time: 10ms. Degrouping voltage (DP) and collision energy (CE) were detailed in supplementary Table S3. The scheduled MRM acquisition method was used, the total scan time was 1.7S, and the MRM detection window was 300S.

3) In the data availability it's stated:

"All data of the study is available from the corresponding authors upon request." This is not desirable for proteomics data and raw files with search output should be made available via a repository like Pride (ProteomeXchange).

Response: The proteomics data has been submitted to Pride (ProteomeXchange) and will become publicly available once the paper is published online.

The data available was as follows: All relevant data supporting the key findings of this study are available within the article and its supplementary information files. The datasets analyzed during the current study are available in PRIDE (PXD041236; PXD041294), TCGA and Human Protein Atlas. Source data are provided with this paper.